# Grain filling in barley relies on developmentally controlled programmed cell death

Volodymyr Radchuk [1✉], Van Tran[1], Alexander Hilo [1], Aleksandra Muszynska [1], Andre Gündel [1], Steffen Wagner [1], Joerg Fuchs[1], Goetz Hensel [1], Stefan Ortleb [1], Eberhard Munz[1], Hardy Rolletschek[1] & Ljudmilla Borisjuk[1✉]

Cereal grains contribute substantially to the human diet. The maternal plant provides the carbohydrate and nitrogen sources deposited in the endosperm, but the basis for their spatial allocation during the grain filling process is obscure. Here, vacuolar processing enzymes have been shown to both mediate programmed cell death (PCD) in the maternal tissues of a barley grain and influence the delivery of assimilate to the endosperm. The proposed centrality of PCD has implications for cereal crop improvement.

[1] Leibniz Institute of Plant Genetics and Crop Plant Research (IPK) Gatersleben, Seeland, Germany. ✉email: radchukv@ipk-gatersleben.de; borisjuk@ipk-gatersleben.de

The accumulation of starch and protein in the developing endosperm of cereals such as barley (*Hordeum vulgare*) and bread wheat (*Triticum aestivum*) depends on the maintenance of a continuous supply of assimilate from the maternal plant. The vasculature through which this material flows, however, reaches only as far as the endosperm's surrounding tissue, which itself deteriorates as the endosperm develops. The terminus of the main vascular bundle linking the developing grain to the maternal tissue is located in a structure referred to as the crease. On the filial side of the crease, the vascular bundle is connected to a complex multicellular structure called the nucellar projection (NP) (Fig. 1a, Supplementary Movie 1). The vascular bundle and the NP provide the main conduit for the movement of assimilate to the endosperm transfer cells (ETCs), which lie against the NP[1]. The transfer process is supported by a number of transporter proteins acting on both the maternal and filial sides[2,3]: sucrose transporter 1 (SUT1) is largely confined to the ETCs[4], whereas both SWEET11a and SWEET11b ("Sugars Will Eventually be Exported Transporter") are expressed in the NP[5].

Within the NP, the product of the gene *Jekyll* conditions programmed cell death (PCD) at a specific developmental stage[6],

while simultaneously, a number of proteases are activated[2,7]. Among these proteases are various vacuolar processing enzymes (VPE), which are required for PCD to occur[8,9]. Plant VPEs are both structurally and functionally similar to animal caspases, but this similarity does not extend to sequence homology. VPEs can act either as a protease, or a peptide ligase[10,11]. Eight VPE proteins are encoded in barley genome[12], but only three of them (*VPE2a*, *VPE2b*, and *VPE2d*) are all actively transcribed in the NP[13,14]. The proteins share between 84.7% and 95.2% identity with one another. The purpose of the present investigation was to elucidate the contribution of these VPEs to the grain-filling process in barley. The experiments were designed to explore the importance of the *VPE2a-VPE2d* genes with respect to both the PCD of the NP and the removal of the cellular barriers obstructing the flow of assimilate from the maternal plant to the developing endosperm.

## Results

**The cellular disintegration of the NP is driven by vacuolar-mediated PCD.** Inspection of electron micrographs of the NP in the developing barley grain revealed a characteristic centripetal

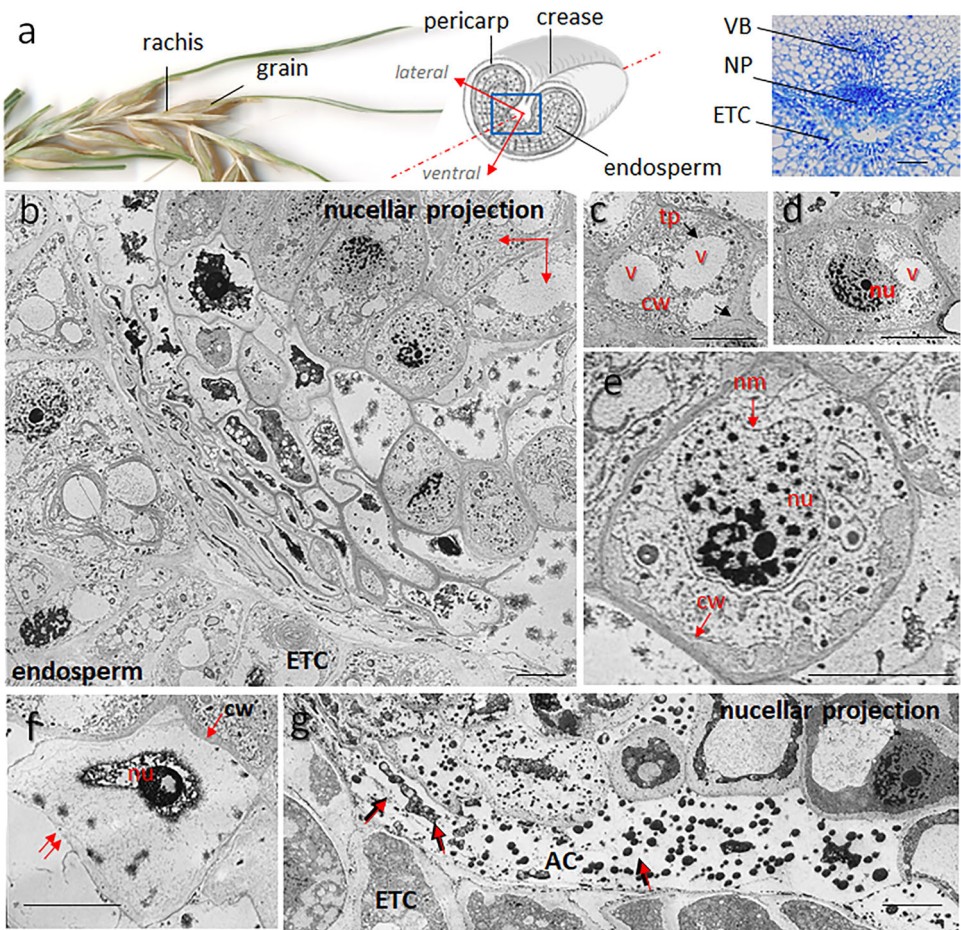

**Fig. 1 Programmed cell death in the nucellar projection of the developing barley grain. a** From left to right: a fragment of the barley spike, the structure of a grain and a close-up of the grain's crease region. The blue square in the scheme shows the location of the microscopic section (right; stained by toluidine blue). The main axes in the scheme define the orientation of the electron microscopic image in **b. b** Electron microscopic image showing the margins of the NP facing the endosperm (9 DAF). **c** An elongating NP cell containing numerous vacuoles. **d** A condensing nucleus. **e** Rupture of the nuclear membrane, a disintegrating nucleus, and the beginning of the clearing of the cytoplasm. **f** Rupture of the cell membrane (double-arrowed), allowing cell remnants to be released into the apoplastic space in front of ETCs. **g** Apoplastic cavity at the NP margins in front of the ETCs. Still disintegrating cells releasing their content into the apoplastic cavity; cell debris is arrowed. Bars = 100 μm in **a** and 15 μm in **b**–**g**. Abbreviations: *AC* apoplastic cavity, *cw* cell wall, *ETC* endosperm transfer cells, *nm* nuclear membrane, *NP* nucellar projection, *nu* nucleus, *tp* tonoplast, *v* vacuole, *VB* vascular bundle.

gradient in cell degradation (Fig. 1b–g; Supplementary Fig. 1). Cells in the medial region of the NP were characterized by their dense cytoplasm containing plentiful endoplasmic reticulum and small vesicles, as is typical for proliferating tissues. As they matured and expanded, the cells typically enclosed a large central vacuole. Its membrane ruptures and smaller vacuoles are formed, marking the start of PCD (Fig. 1c). Already at this stage, the nucleus had begun to condense (Fig. 1d). Cell disintegration proceeded with the rupture of the nuclear envelope (Fig. 1e), then the collapse of nuclei, finally resulting in the release of their contents into the cytoplasm (Fig. 1f). The disappearance of the inner membrane structures and the formation of irregular clumps are indicative of the degradation of both the cytoplasm and the nucleus, ending in the clearance of the cytoplasm (Fig. 1f). Subsequently, the cell membrane itself ruptured, spilling cell remnants and the degraded cell contents into the apoplastic space in close vicinity to the ETCs (Fig. 1g). This pattern of cellular disintegration is consistent with vacuole-mediated PCD[15,16]. In conclusion, the barley NP undergoes PCD with cell elimination at the boundary to the endosperm.

**Multiple copies of VPE2 gene subfamily are a feature of Triticeae genomes.** Putative VPE polypeptide sequences were identified from a screen of the genome sequences of barley, *Aegilops speltoides* and bread wheat (all of which are members of the *Triticeae* tribe), and were then compared with homologs encoded by the grass *Brachypodium distachyon* (a close relative of the *Triticeae*) and by the more distantly related rice (*Oryza sativa*) and maize (*Zea mays*) genomes (Fig. 2a). The resulting clade structure suggests that genes encoding the endosperm-specific barley VPE1, the pericarp-specific barley VPE4, and VPE3/VPE5 (expressed in vegetative tissue) arose prior to the diversification of the grasses. (Note that owing to its hexaploid status, the bread wheat genome includes triplicate subfamily members.) With respect to the VPE2 subfamily, in contrast, although only a single member is present in the maize, *B. distachyon* and rice genomes, the barley genome harbors four copies (*VPE2a* through *VPE2d*), as does that of *Ae. speltoides* and each of the three sub-genomes of bread wheat. The implication is that the expansion of VPE2 subfamily occurred in a common ancestor before separation of the Triticeae tribe. The Triticeae *VPE2c* sequences cluster with its orthologs from the non-Triticeae species, suggesting their ancestral state.

**The effect on key grain traits of suppressing VPE2 subfamily genes.** Although barley *VPE3*, *VPE4,* and *VPE5* genes were expressed in a number of vegetative and regenerative tissues (Supplementary Fig. 2a), the genes of the VPE2 subfamily were abundantly transcribed only in generative tissues. All four VPE2 genes were active in the anthers, whereas the transcription of *VPE2a*, *VPE2b*, and *VPE2d* was detectable in the developing caryopsis (Fig. 2b). The *VPE1*, *VPE3,* and *VPE4* genes were also expressed in the developing caryopsis, but only negligible in the NP (Supplementary Fig. 2b). The VPE1 was shown to have its predominant expression in the late endosperm[13,14], whereas VPE4 was highly active in the pericarp[17]. *VPE2a*, *VPE2b,* and *VPE2d* have been shown to share a similar transcriptional profile in the NP[14,18]. A set of 22 independent RNAi transgenic lines, engineered to express a portion of *VPE2a* RNAi-fragment driven by the gene's native promoter, was generated, from which four homozygous lines with single T-DNA copy were selected: VPE2i-11 and VPE2i-19 with strong phenotypic effects, VPE2i-16 showing the moderate ones and VPE2i-5 with weak phenotype. Because the selected RNAi-fragment of *VPE2a* gene was ~93% identical to the corresponding regions of *VPE2b* and *VPE2d*, the suppression of not just *VPE2a*, but also *VPE2b* and *VPE2d* was

successfully achieved, when analyzing $T_3$ generation of the transgenic plants (Fig. 3a). VPE2b has been shown to exhibit caspase-1-like activity[12]. When crude protein extracts from both transgenic and wild-type (WT) grains were assessed, caspase-1-like activity was clearly lower in the transgenic grains than in WT; furthermore, the level of caspase-3-like, caspase-6-like, and caspase-8-like activity were also lower (Fig. 3b). There was no evidence of any phenotypic effect of the VPE2 subfamily down-regulation prior to anthesis, but once the grain-filling phase began, the transgenic grains became distinct from the WT ones, appearing both thinner and longer. When magnetic resonance imaging (MRI) was used to both visualize the three-dimensional structure of the grains and to estimate the volume occupied by the various tissues (Fig. 3c, d, Supplementary Movies 2 and 3), the analysis revealed that the volume of the transgenic grains' endosperm, the embryo, and the apoplastic cavity was significantly smaller than that of the WT grain, and that the relative volume occupied by the pericarp was increased (Fig. 3e). The transgenic grains accumulated fresh weight more slowly than the WT grains, and the plants set fewer grains per spike (Supplementary Fig. 3a, b). At maturity, whereas the transgenic grains were longer than WT ones, their thousand-grain weight was as much as 29% lower (Fig. 3f). Although the absolute accumulation of starch, protein, and lipid was decreased in the transgenic grains (Supplementary Fig. 3b), the relative representation of these storage compounds did not differ between the grain set by WT and the transgenic plants.

**Suppressing VPE2 subfamily induces aberrant PCD in the NP.** A detailed histological analysis of the NP in grains where VPE2 subfamily was suppressed revealed that, whereas some cell disintegration took place at the periphery, clearance of the cytoplasm, and the complete digestion of the cells did not occur (Fig. 4a). The consequence was that non-eliminated cells accumulated in the apoplastic cavity of the transgenic grains. A terminal deoxynucleotidyl transferase (TdT) dUTP nick-end labeling (TUNEL) assay identified the presence of numerous disintegrating nuclei in NP of the WT grains but only a few disintegrating nuclei in those of the transgenic (Fig. 4a; Supplementary Fig. 4). When flow cytometry was employed to estimate cell numbers in both the maternal and the filial tissue, it was observed that starting from 6 days post anthesis, the transgenic grains harbored a higher number of cells in the maternal seed tissues (Fig. 4c). The bulk of the maternal tissues consists of the pericarp and the nucellus/NP[13]. Upregulated expression of *VPE4*, predominantly located in the pericarp and known to eliminate cells via PCD[17], was detected during mid phase of grain development (Supplementary Fig. 5). This is expected to counterbalance the elevated number of maternal cells owing to the VPE2 repression. Therefore, a higher cell number detected in the maternal seed tissues was presumably because fewer cells had been exposed to PCD in the NP.

A comparable number of cells was present in the endosperm of the transgenic and WT grain during the early phase of grain development, but the numbers diverged in favor of the WT as grain-filling began (Fig. 4c). Given that the product of *Jekyll1* regulates PCD in the NP[6] and that cathepsin B possesses caspase-3-like activity required for PCD[19], it was of interest to quantify the abundance of *Jekyll1* and *cathepsin B1* transcripts: the *Jekyll* gene was less transcribed at early development (6 and 9 DAF) while *cathepsin B1* transcripts were decreased during later stages (9 and 12 DAF) in the transgenic grains (Fig. 4d). As the growth of the endosperm depends on the establishment of functional ETCs, a comparison was drawn between the ETCs present in the transgenic and WT grains; while the WT ETCs exhibited a dense,

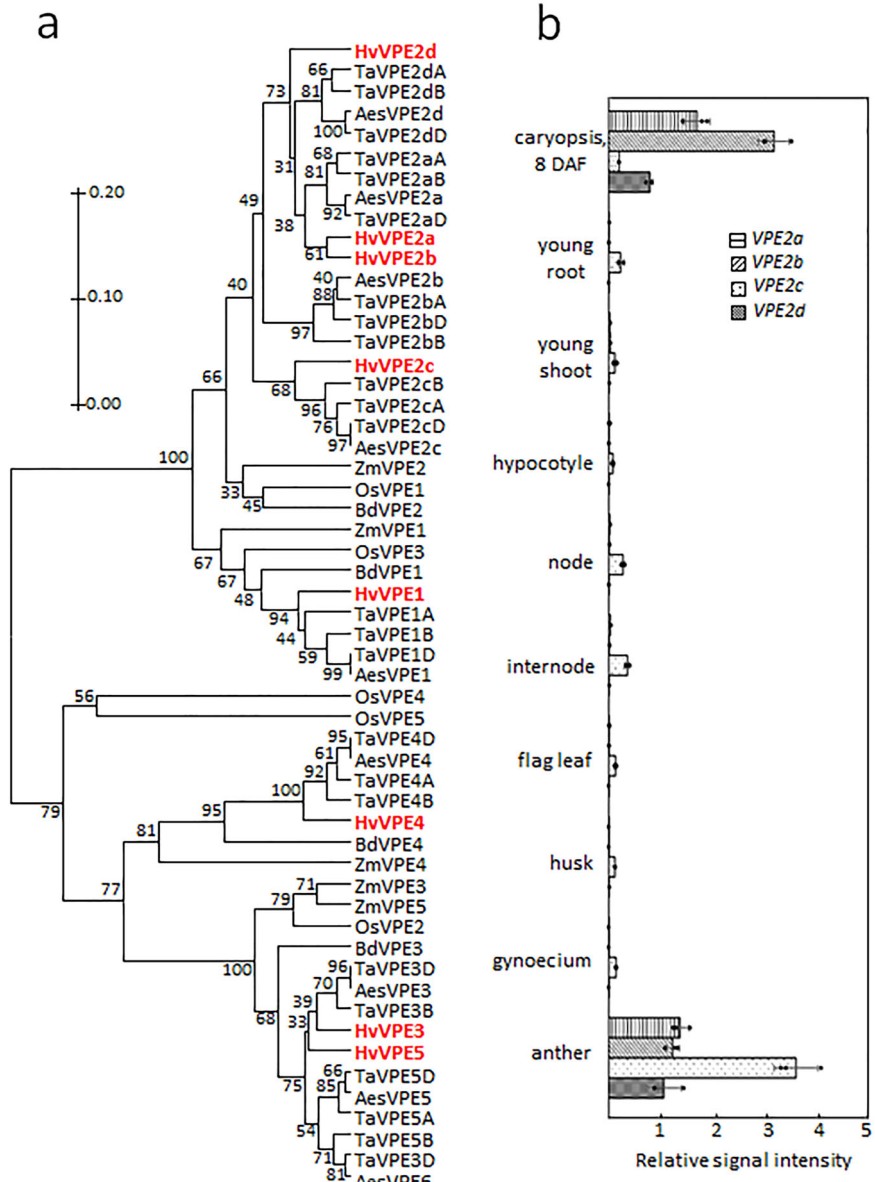

**Fig. 2 The VPE2 gene subfamily expansion in the Triticeae and its expression in the barley plant. a** A bootstrap consensus tree of the VPE protein sequences from *Hordeum vulgare* (Hv), *Aegilops speltoides* (Aes), *Triticum aestivum* (Ta) (all belong to Triticeae tribe), *Zea mays* (Zm) (Andropogoneae), *Oryza sativa* (Os) (Oryzeae), and *Brachypodium distachyon* (Bd) (Brachypodieae). The maximum-likelihood tree was supported by 1000 bootstrap replicates. The scale bar represents evolutionary distances, as quantified by the number of substitutions per amino-acid residue. The barley VPEs are shown in red. Gene IDs are given in Supplementary Table 4. **b** Transcript abundances of *VPE2a*, *VPE2b*, *VPE2c*, and *VPE2d* in various parts of the barley plant, assessed using qRT-PCR. Data given as the mean ± SD (*n* = 3). Abbreviation: *DAF* days after flowering.

organelle-rich cytoplasm, formed rather small vacuoles and developed thick cell walls, many of the ETCs in the transgenic grains contained either a large central vacuole (line VPE2i-11) or numerous smaller ones (line VPE2i-19), and formed thin cell walls (Fig. 4b; Supplementary Fig. 6). Consistent with this difference, the abundance of transcripts of both *END1* and *BETL*, the products of which are associated with the differentiation of the ETCs[2,20], was lower at most time points in the transgenic than in the WT grains (Fig. 4e). The overall conclusions were that, first, VPE activity is relevant for PCD in the NP, and second, the suppression of VPE2 subfamily disrupted not only the NP but also the differentiation of the ETCs.

**A defective assimilate transfer route inhibits sucrose transport into the endosperm.** The kinetics of sucrose uptake and its

distribution in the developing grains set by WT and transgenic plants was explored using $^{13}C/^{1}H$ MRI[1]. In WT grains, the signal was first detected in the apoplastic cavity just in front of the ETCs, consistent with a rapid transport of sucrose into the endosperm; in contrast, in the transgenic grains, the earliest signal was recovered from the rachis, reaching a higher level in the NP only somewhat later (Fig. 5a). The abundance of transcript produced by genes encoding the major sucrose transporter proteins during the course of grain filling was subsequently measured. *SUT1*, a gene induced by high levels of the substrate and predominantly expressed in the ETCs[4], was substantially downregulated in the transgenic grains; *SUT2*, a gene involved in the maintenance of sucrose balance in the vacuole[21], was unaffected by the presence of the transgene; meanwhile, both *SWEET11a* and *SWEET11b* (encoding sucrose exporters and expressed mostly in the NP[5]) were both upregulated in the

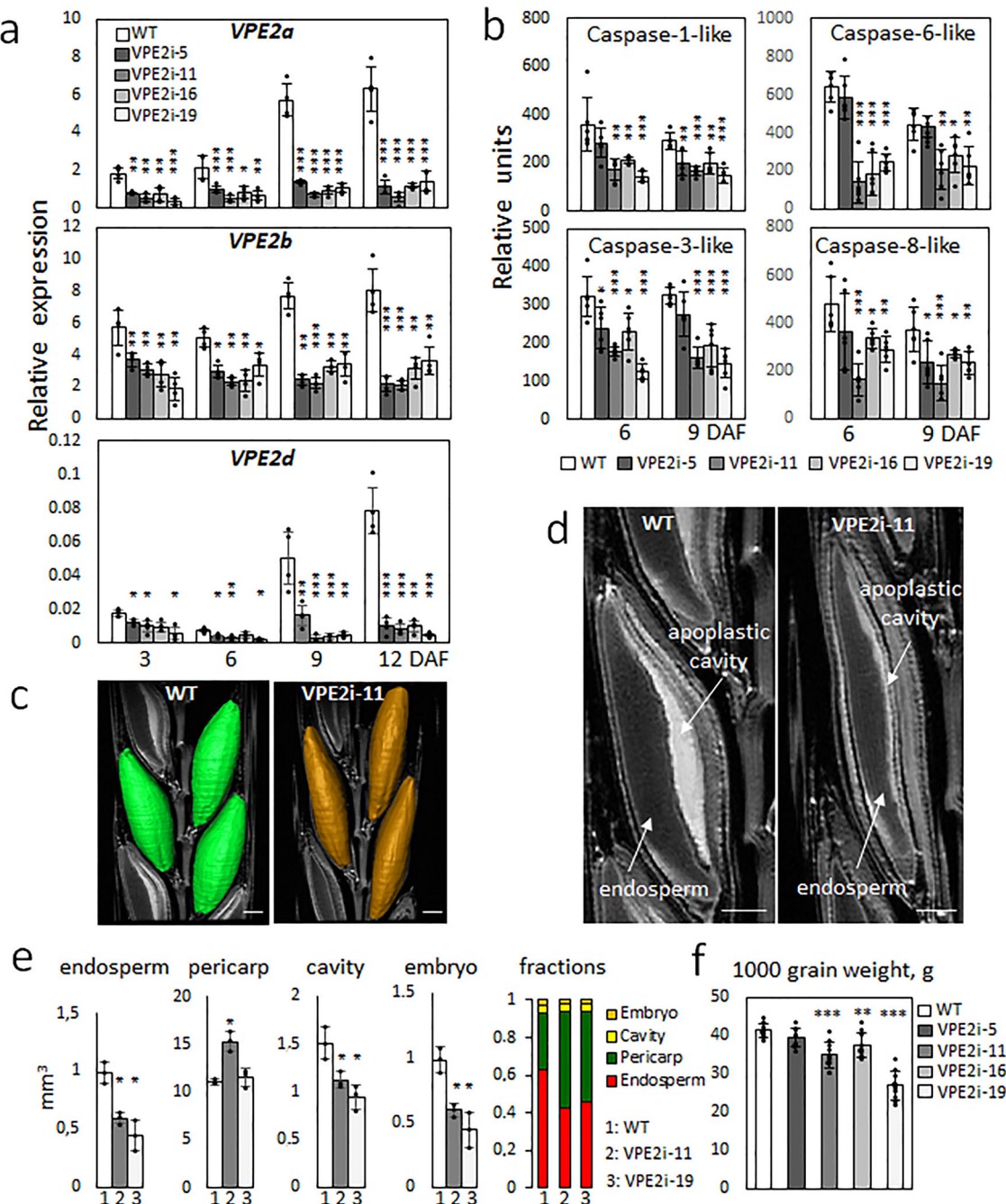

**Fig. 3 The phenotypic effect of suppressing VPE2 subfamily. a** The abundance of *VPE2a*, *VPE2b*, and *VPE2d* transcripts in the developing grains of WT and RNAi-VPE2 transgenic lines, as measured by qRT-PCR. **b** Caspase-like activities in the developing grains of WT and transgenic lines. **c** MRI-derived three-dimensional models of the WT (labeled green) and VPE2i-11 (brown) caryopses (nine DAF). **d** The inner structure of the developing living caryopses, as acquired by MRI (9 DAF). Three-dimensional visualization of WT, VPE2i-11, and VPE2i-19 grains is shown in Supplementary Movies 1, 2, and 3. **e** The volume of each of the various components of the caryopsis and their relative contribution to the grain's overall volume. **f** The weight of 1000 mature grains. Bars = 1 mm in **c** and **d**. Data given as the mean ± SD ($n = 4$ in **a**, $= 6$ in **b**, $= 3$ in **e**, and $= 10$ in **f**). *, **, ***: means differ at, respectively, $P < 0.05$, $< 0.01$, and $< 0.001$. Abbreviation: *DAF* days after flowering.

transgenic grains (Fig. 5b). Thus, reduced intensity of PCD in the NP disrupts assimilate transport from the maternal to the filial tissue, leading to increased sucrose accumulation in a spike and maternal parts of a grain.

**Starch synthesis in the grain is altered by the downregulation of VPE2 subfamily expression.** A comparison of the composition of grains set by WT and RNAi-VPE2 transgenic plants revealed

the latters' enhanced accumulation of both sucrose and hexose, and the signaling compound trehalose 6-phosphate (Fig. 6a). The content of both hexose phosphates (the cleavage products of sucrose) and ADP-glucose (a precursor of starch) was depleted throughout the development of the transgenic grains. The rate of accumulation of starch was slower in the transgenic grains throughout the grain-filling period (Fig. 6b). At the transcriptional level, the activity of starch synthesis-related genes encoding an ADP-glucose transporter, ADP-glucose pyrophosphorylase

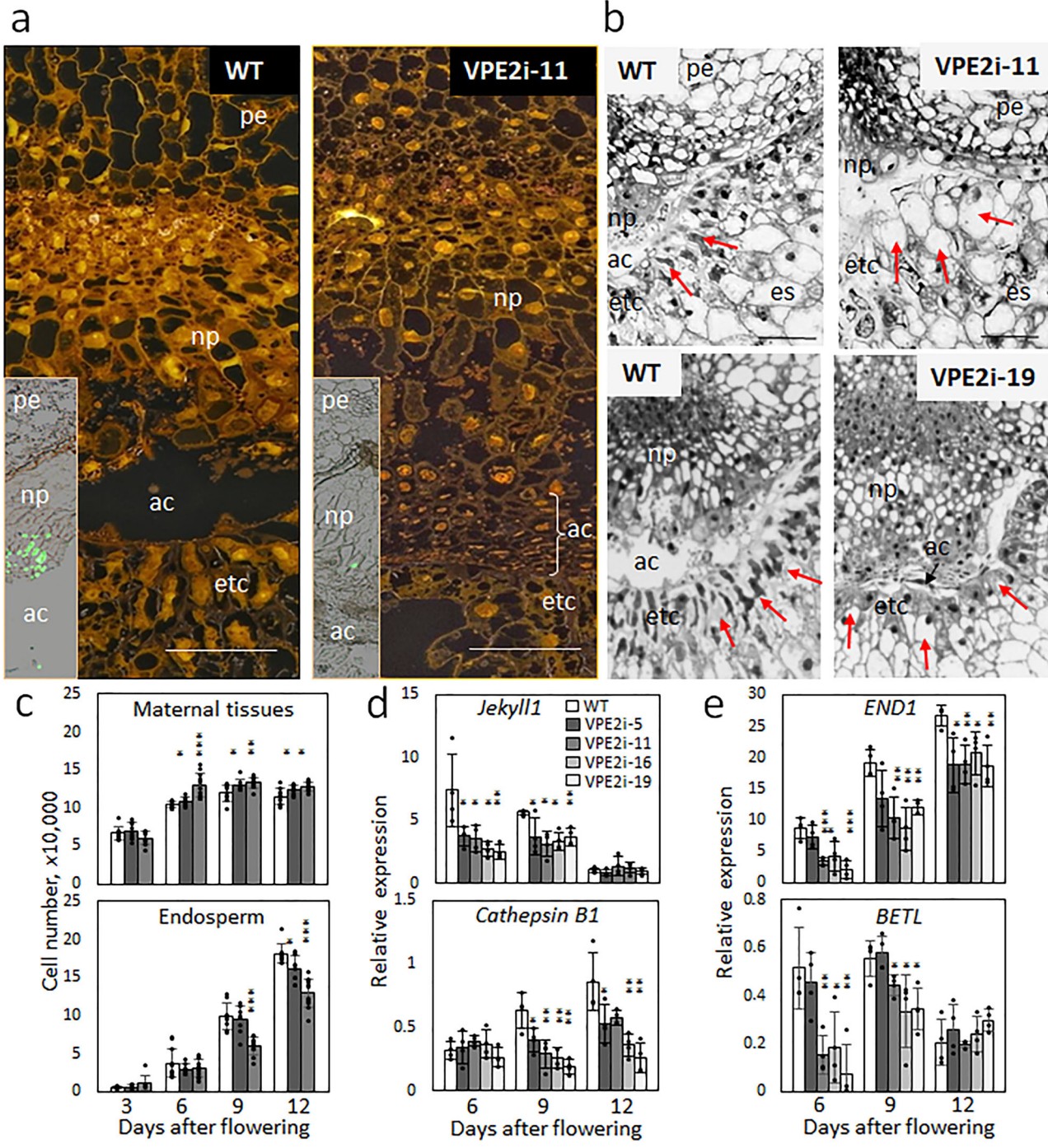

**Fig. 4 The effect of suppressing VPE2 subfamily on development at the maternal–filial interface. a** The characteristic of cell differentiation in NP of a WT (left panel) compared with VPE2i-11 grain (right panel) at 9 DAF. Non-disintegrating cells filled the apoplastic cavity of the transgenic grain. Inlets visualize nuclei degradation (TUNEL assay) in NP of WT and VPE2i-11 grains (see also Supplementary Fig. 4.) **b** The morphology of the ETCs, formed in a WT grain (left panel), is characterized by densely packed cells (arrowed), whereas the ETCs formed in a transgenic grain (right panel) are highly vacuolated. **c** Cell numbers in the maternal tissue and the endosperm in developing grains of WT and RNAi-VPE2 transgenic lines. **d** and **e** Transcript abundances of PCD-related genes *Jekyll1* and *cathepsin B1* (**d**) as well as *END1* and *BETL*, both encoding proteins involved in the ETC differentiation (**e**), in the developing grains of WT and transgenic lines. Data shown as the mean ± SD ($n = 10$ in **c**, $= 4$ in **d** and **e**). *, **, ***: means differ at, respectively, $P < 0.05$, $< 0.01$, and $< 0.001$. Bars = 50 mm in **a** and **b**. Abbreviations: *ac* apoplastic cavity, *DAF* days after flowering, *es* endosperm, *etc* endosperm transfer cells, *np* nucellar projection, *pe* pericarp, *WT* wild type.

subunit S1a (*AGP-S1a*) and granule-bound starch synthase 1a (*GBSS1a*), all of them are active exclusively in the endosperm[22], was lower in the transgenic grains than in the WT, whereas the opposite was the case for a pericarp-specific[22] *GBSS1b* (Fig. 6c). An analysis of grain sections, based on Fourier-transform infrared micro-spectroscopy[23], revealed that the transgenic grains' pericarp accumulated more sucrose and starch than did the equivalent tissue in the WT grains (Fig. 6d, e). The overall conclusion was that the compromised allocation of sucrose resulting from knocking down VPE2 subfamily resulted in an altered profile of

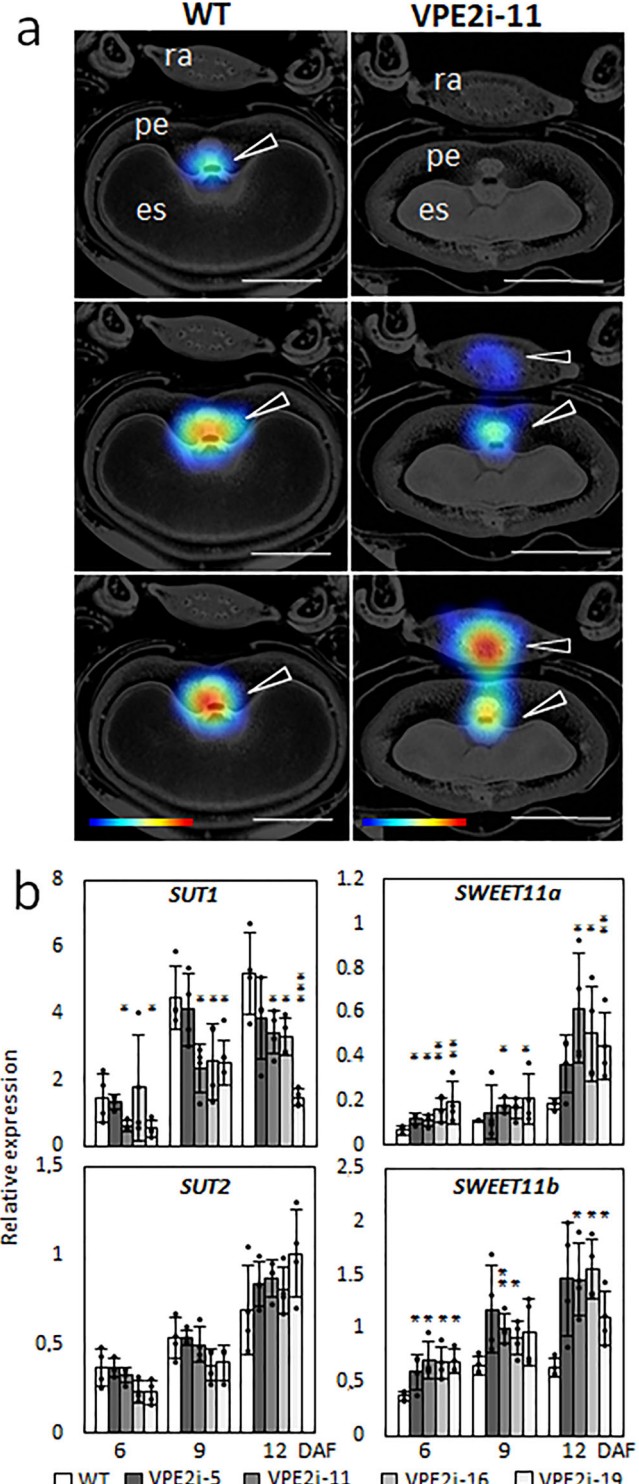

**Fig. 5 The suppression of VPE2 subfamily alters the allocation of sucrose in the grain. a** The distribution of $^{13}$C labeled sucrose, as captured by MRI after 1 (above), 3 (middle), and 8 h (bottom) at 9 DAF. The sucrose concentration is indicated by the color-coded bar (below) with max (min) values in red (blue). **b** The abundancies of *SUT1*, *SUT2*, *SWEET11a*, and *SWEET11b* transcripts in the developing grains of WT and RNAi-VPE2 transgenic lines. Data shown as the mean ± SD (*n* = 4). *, **, ***: means differ at, respectively, *P* < 0.05, < 0.01, and < 0.001. Bars = 1 mm in **a**. Abbreviations: *DAF* days after flowering, *es* endosperm, *pe* pericarp, *ra* rachis.

carbohydrate storage in the grain: since more of the sucrose was retained within the maternal portion of the grain, less was available for starch synthesis in the endosperm.

**The effect of suppressing VPE2 subfamily on the grains' nitrogen and central metabolism**. The uptake into, and distribution within the grain of the major transport amino-acid glutamine (Gln) was assessed by feeding plants with $^{15}$N-Gln. The RNAi repression of the VPE2 subfamily altered the label distribution within the grain: a greater proportion of the $^{15}$N label accumulated in pericarp of the transgenic grain rather than in the WT pericarp (Fig. 7a). Expression of genes encoding three hordeins, amino-acid permease 3 (AAP3), and cation amino-acid transporter 1 (CAT1) were all downregulated in the transgenic grains (Fig. 7b, Supplementary Fig. 5). The steady-state level of free amino acids in the transgenic grains was ~80% of that in the WT grain (Supplementary Data 1), driven mostly by their reduced accumulation of alanine.

A comprehensive metabolic analysis of developing grains sampled during their transition from the early to the main grain-filling stage was based on high-resolution mass spectrometry (MS) (Fig. 7c; Supplementary Data 1). In general, with respect to metabolites associated with glycolysis, the tricarboxylic acid cycle, ascorbate, and nucleotide metabolism, the pentose phosphate pathway, the Calvin cycle, and amino-acid metabolism, a lower steady-state level pertained in the transgenic grain, suggestive of a repression in the central metabolism. On the other hand, the level of glutathione-related metabolites involved in redox regulation, pyrophosphate (an alternative energy donor), and the phenyl-propanoid precursors quinic acid and shikimate was enhanced, as were those of 2P-glycolate and glycerate (intermediates of photorespiration, restricted to the pericarp chlorenchyma). The levels of 1-aminocyclopropane-1-carboxylic acid (ethylene precursor) and panthothenate (cofactor synthesis) were also higher in the transgenic than in the WT grains. Overall, the picture was one in which one of the consequences of compromising the delivery of assimilate as a result of the loss of VPE2 activity was a wide-ranging adjustment to the grains' central metabolism.

## Discussion
PCD is an integral part of the life cycle of all multicellular organisms. However, the molecular basis of this process does differ between plants and animals[16,24]. Although in animal cells, PCD is executed by diverse cytosolic caspases, their orthologs have not been identified in plant genomes; instead, the evidence suggests that VPEs are key to control PCD in plants[8].

Here, it has been demonstrated that the PCD, which occurs within NP of the barley grains is mediated by vacuole, and that VPEs are an essential component of the process. The repression of *VPE2a-VPE2d* genes accomplished via RNAi resulted in a disturbance to the wild-type progression of PCD, thereby compromising grain filling by delaying the transport of sucrose into the developing endosperm. In grain set by RNAi transgenic plants, although the endosperm remained the major sink for sugar, sucrose was also accumulated within the maternal pericarp. A similar disruption was noted with respect to nitrogenous compounds. Reducing the delivery of nutrients to the endosperm has been shown to decrease metabolic flux without altering the pathway utilization patterns[25]. As a result, starch and protein accumulated in the pericarp at the expense of the endosperm, a structure that normally acts as the major site for the deposition of storage compounds. The transgenic effect on grain weight was rather mild owing to not only moderate downregulation of VPE2

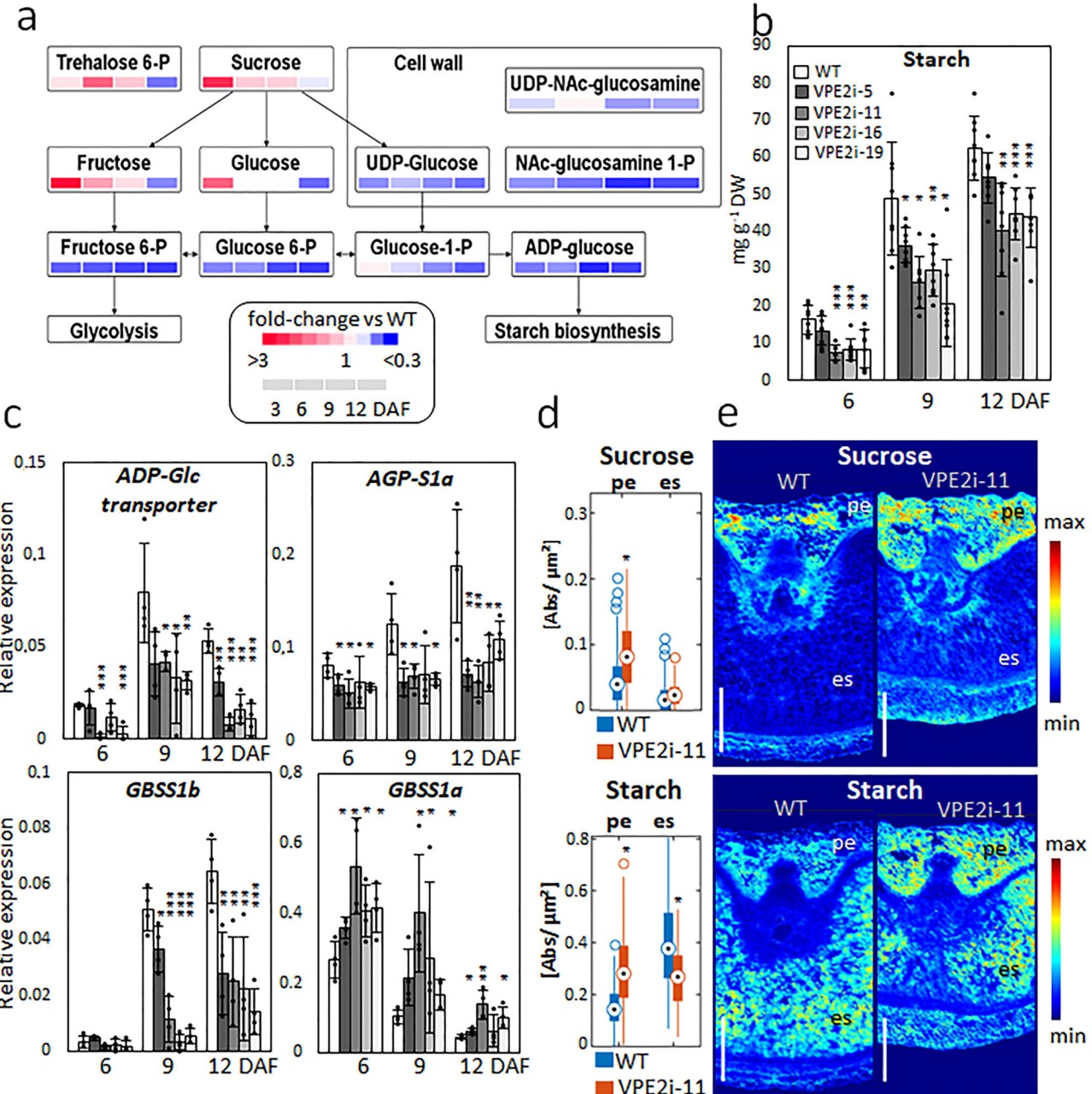

**Fig. 6 The suppression of the VPE2 subfamily compromises the accumulation of starch in the endosperm. a** A metabolite heat map illustrating the abundance of various sugars in the developing grain. **b** Temporal profile of starch accumulation in WT and RNAi-VPE2 transgenic lines. **c** The abundance of transcripts generated from genes encoding an ADP-glucose transporter (*AGP-Glc transporter*), small subunit S1a of AGPase (*AGP-S1a*), and granule-bound starch synthases 1a (*GBSS1a*) and 1b (*GBSS1b*) in the grain of WT and transgenic lines. **d** Comparative Fourier-transform infrared (FTIR) analysis of sucrose and starch in pericarp (PE) and endosperm (ES) of WT and transgenic grains measured at 9 DAF. Data are given in relative units (dot point, median; box, interquartile range [IQR]; lines, 1.5 × IQR; stars indicate statistical significance at $P < 0.05$ by the Mann–Whitney $U$ test). **e** Representative FTIR-based images of sucrose (upper panels) and starch (lower panels) in WT and transgenic grains. The concentrations are color-coded. Bars = 500 μm. Data shown as the mean ± SD ($n = 8$ in **a** and **b**, = 4 in **c**, = 5 in **d**). *, **, ***: means differ at, respectively, $P < 0.05$, < 0.01, and < 0.001. Abbreviations: *DAF* days after flowering, *es* endosperm, *pe* pericarp, *WT* wild type.

genes but also transcriptional activation of *SWEET11a* and *SWEET11b*, leading to a compensatory stimulation of sucrose delivery toward endosperm.

Unlike the situation in rice grains[26,27], where assimilate is transported from the maternal into the filial portion of the developing grain via both the nucellar epidermis as well as NP and ETCs, in barley, the NP together with the ETCs represent the only conduit of assimilate[1]. The formation of NP in the *Triticeae* progenitor is associated with the expansion of VPE2 subfamily

and the evolution of *Jekyll*[28]. The effect of these evolutionary events is a broadening of the maternal post-vascular route, which supports a more effective flow of nutrients into the endosperm.

During the process of sexual reproduction in plants, PCD is used to remove specific cells and tissues in order to provide space for the rapidly expanding filial structures[17]. Its contribution to feeding the zygote and endosperm is to remobilize the nutrients tied up in redundant cells[29,30]. The present investigation has shown that PCD at maternal–filial borders of the barley grains

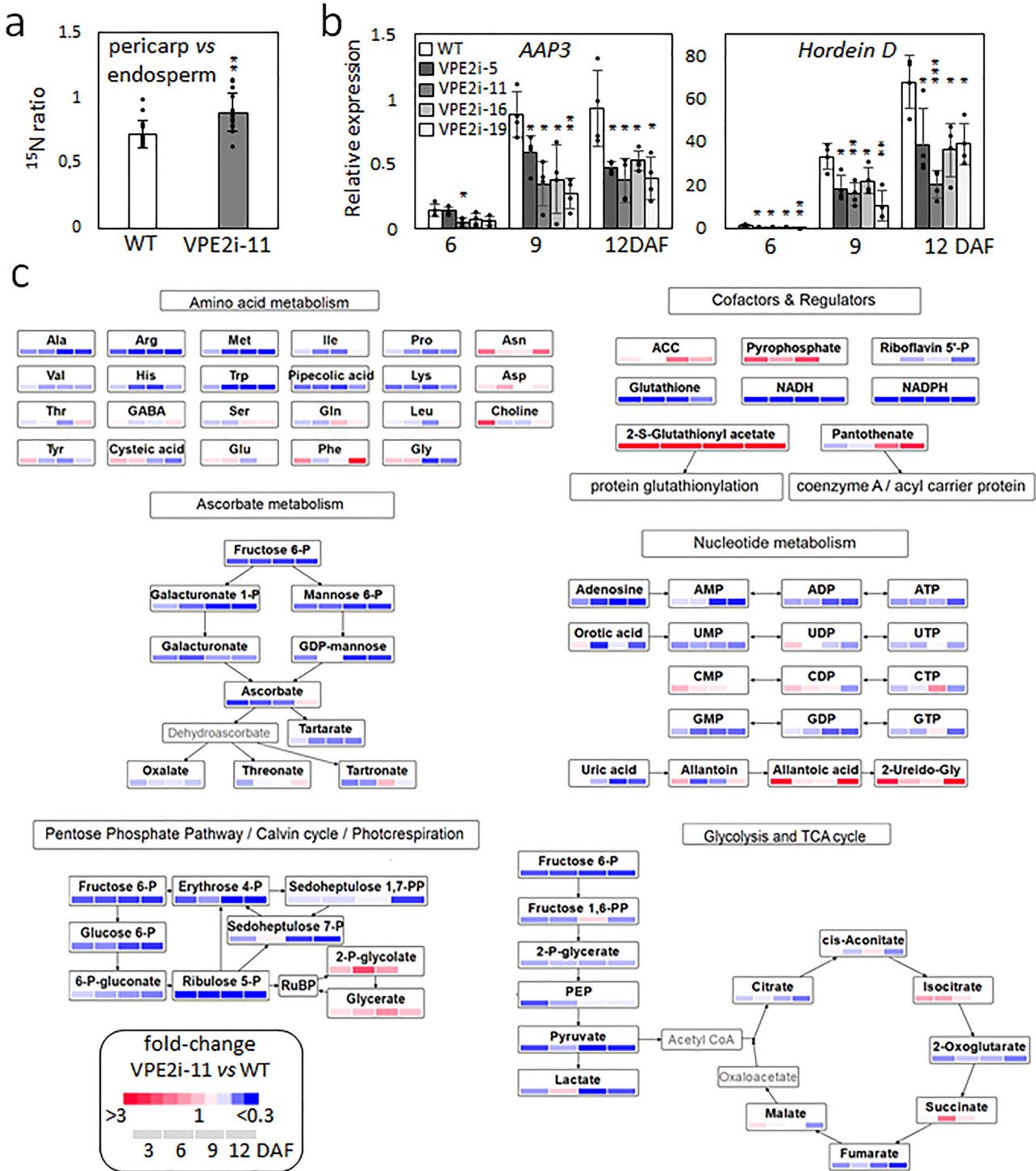

**Fig. 7 The metabolic effect of suppressing VPE2 subfamily. a** The ratio of $^{15}$N accumulated in pericarp versus endosperm as indicator of differential nitrogen allocation within a grain (9 DAF). **b** The abundances of transcripts generated from genes encoding amino-acid permease 3 (AAP3) and hordein D. **c** A heat map illustrating the abundance of various amino acids, co-factors, and regulators, components of ascorbate and nucleotide metabolism, components of the pentose phosphate pathway, the Calvin cycle and photorespiration, and of glycolysis and the tricarboxylic acid cycle (TCA) measured using LC/MS in immature grains of the transgenic line VPE2i-11 versus WT. Concentrations indicated by color coding. Data are shown as the mean ± SD ($n = 15$ in **a**, $= 4$ in **b**, $= 8$ in **c**). *, **, ***: means differ at $P < 0.05$, $< 0.01$, and $< 0.001$, respectively. Abbreviations: *DAF* days after flowering, *WT* wild type.

also serves to widen the rather narrow post-vascular assimilate route, thereby accelerating assimilate allocation. To achieve this, the NP is dismantled, starting at its outer surface, until the entire content has been released into an apoplastic space in front of the ETCs. By disintegrating its cells, the NP gets more permeable to assimilate flow/transport. As a result, a major barrier to the

transport of assimilate is removed, allowing the endosperm to drive the process of grain filling.

The nucellar tissues are an important link in the nutrient allocation chain toward the fertilization product. Its morphology and mode of action can differ in seeds of distinct species, and engages distinct mechanisms[30,31]. PCD in the NP is an important

control element for the post-vascular nutrient transport route in barley grains. Manipulating the timing of this PCD could represent a novel means of increasing the efficiency of the grain-filling process in the major cereal crop species.

## Methods

**Plant material and growth conditions**. Wild-type (WT) and the transgenic barley (*H. vulgare* L.) plants were grown as previously described[13]. Plants were divided into six sub-batches, and grains of separate sub-batches were collected at the defined developmental stages and frozen in liquid nitrogen. Accumulation of fresh weight was calculated from c. 50 developing caryopses collected from c. 10 spikes. Thousand-grain weight and other seed parameters were calculated using the Marvin seed analyzer (GTA Sensorik).

**Plasmid construction and plant transformation**. The RNAi construct pVPE2i consisted of the *VPE2a* promoter region (1061 nucleotides (nt) upstream of the ATG start codon), a sense *VPE2a* (gene ID HORVU2Hr1G091880.1) fragment (554 nt), an intron of *gibberelic acid 20 oxidase* from potato (*Solanum tuberosum* L.) (199 nt) and an antisense *VPE2a* fragment (554 nt). The *VPE2a* fragment, selected for RNAi, shares 92.2% identity to the *VPE2b* and 91.1% identity to *VPE2d*, correspondingly. The appropriate DNA fragments were PCR amplified and cloned, using the specific restriction sites, into a modified pAX vector that already contained the potato intron and the nopaline synthase terminator. The primers and restriction sites were as follows: for the *VPE2a* promoter, 5′-TT<u>AGGCCT</u>GATTATCGAGACTTTCCAT-3′ (*Stu*I underlined, as are further restriction sites) and 5′-GAT<u>ACTAGT</u>TCATGTGCG TGTGCTAGAGTG-3′ (*Spe*I); for the *VPE2a* sense fragment, 5′-GAT<u>GGCTGCAG</u> GGTGGCTTTG-3′ (*Pst*I) and 5′- GCTTTTGCCGTA-<u>ACTAGT</u>TGATG<u>AAGTCG</u>-3′ (*Spe*I); for the *VPE2a* antisense fragment, 5′-CTAG<u>CACACGC</u>ACATGAA <u>GTCGAC</u>G-3′ (*Sal*I), and 5′-TT<u>GGATCC</u>ATGCTTTTGCCG-TAACAC-3′ (*Bam*HI). The VPE2i cassette was sequenced (LGC Genomics) to confirm the sequences and orientation of the *VPE2a* fragments. Then, the cassette was introduced into the p6U binary vector (DNA Cloning Service) with *Sfi*I. RNAi transgenic barley lines were generated by *Agrobacterium tumefaciens*-mediated transformation of immature embryos of cv Golden Promise as described[32].

**RNA extraction and reverse transcription-qPCR**. Total RNA was isolated using Trizol reagent (ThermoFisher Scientific), treated with DNase (Qiagen), and cleaned with RNeasy kit (Qiagen). The extracted total RNA was reverse-transcribed into single-stranded cDNA using QuantiTect Reverse Transcription kit (Qiagen). Quantitative reverse transcription-polymerase chain reactions (qRT-PCRs) were performed with QuantStudio 6 Flex Real-Time PCR system (Applied Biosystems) using PowerUp SYBR Green Master Mix (Applied Biosystems). The relative mRNA levels were determined using the $\Delta\Delta CT$ method and normalized to that of *H. vulgare* actin gene (HORVU1Hr1G002840). Primers are listed in Supplementary Table 2. Experiments were run mostly with four biological replications and three technical repetitions each.

**Caspase assay**. The samples for caspase assays were homogenized in liquid $N_2$ and re-suspended in 2× CASPB buffer (100 mM HEPES, 0.1% CHAPS, 1 M DTT, pH 7.0) at 4°C. Cell debris was separated by centrifugation at 13,000 rpm for 10 min at 4°C. Protein concentration in the extracts was estimated by Bradford assay (BioRad). Caspase-like activities were measured in 150 µl reaction mixtures containing 25 µg of protein sample and 10 µM of caspase substrate. Caspase-like activities were detected using the following substrates: acetyl-Tyr-Val-Ala-Asp-7-amido-4-methyl coumarin (Ac-YVAD-AMC) for caspase-1 activity, acetyl-Asp-Glu-Val-Asp-7-amido-4-methyl coumarin (Ac-DEVD-AMC) for caspase-3 activity, acetyl-Val-Glu-Ile-Asp-7-amido-4-methyl coumarin (Ac-VEID-AMC) for caspase-6 activity, and acetyl-Ile-Glu-Thr-Asp-7-amido-4-methyl coumarin (Ac-IETD-AMD) for caspase-8 activity. Emitted fluorescence was measured after 1-hour incubation at room temperature with a 360 nm excitation wavelength filter and 460 nm emission wavelength filter in a spectrofluorometer (Spectra Max Gemini). Six biological replications with three technical repetitions each were performed for the determination of each value and standard deviations were calculated. The system was calibrated with known amounts of AMC hydrolysis product in a standard reaction mixture. Blanks were used to account for the spontaneous breakdown of the substrates.

**Flow cytometric analysis**. Grains with detached embryos (12 grains per stage and line) were subjected to absolute cell counting using flow cytometry. For this, nuclei were isolated as described[33] using nuclear isolation buffer[34] supplemented with 50 µg ml⁻¹ propidium iodide and 50 µg ml⁻¹ DNase-free RNase. Nuclei were analyzed and counted on a CyFlow Space flow cytometer (Sysmex Europe).

**Histochemical techniques**. For light microscopy, the cross-sections (2 mm) of barley caryopses were fixed in 2% (v/v) glutaraldehyde and 2% (v/v) formaldehyde in 50 mm cacodylate buffer, pH 7.2, overnight at 8°C. After washing, samples were dehydrated in a graded ethanol series following by embedding in Spurr's low

viscosity resin (Sigma Aldrich). Samples were sectioned (1 µm) on a Reichert-Jung Ultracut S microtome (Leica), stained with 2% crystal violet (w/v), and examined with a Zeiss Axio Imager light microscope (Carl Zeiss).

For electron microscopy, Spurr embedded probes were sectioned at 70 nm, collected on 75 mesh copper grids, and stained with uranylacetate and lead citrate[35]. Samples were examined using Tecnai20 electron microscope (FEI) at 120 kV. Digital recordings were made with a Megaview III (Soft Imaging Systems).

**TUNEL assay**. WT and transgenic grains were embedded in Paraplast Plus (Leica) and sectioned into 12-µm-thick cross-sections by a rotary microtome (Leica RM 2165). TUNEL assay (Roche) was used to label blunt ends of double-stranded DNA breaks in situ. Both, the negative (without TdT) and positive (treatment with 50 U/ml of DNase I, Roche) controls were included. Afterwards, the sections were rinsed three times in PBS and closed in Prolong Diamond Antifade (Thermo Fischer Scientific) to investigate under a Zeiss Imager microscope using fluorescence (excitation wavelength 488 nm) and bright field. The outcome of the TUNEL assay was documented in AxioVision (Zeiss).

**FTIR imaging and data processing**. Samples were frozen in liquid nitrogen and embedded in Tissue-Tek cryomolds using Tissue-Tek O.C.T. (Sakura Finetek) at −20°C. Embedded tissues were cross-sectioned (16 µm) with a cryotome CryoStar NX7 (Thermo Fisher Scientific) and transferred onto MMI membrane slides (Molecular Machines & Industries). Tissue sections were lyophilized and stored in darkness at room temperature until analysis. These slides also were used for internal standardization as described below.

Imaging was performed using a Hyperion 3000 FTIR microscope (Bruker Optics) coupled to a Tensor 27 FTIR spectrometer (Bruker Optics) with an internal mid-infrared source. The focal plane array detector (64 × 64 pixel) was used in transmission mode. The imaging system was purged with dry air continuously. FTIR images were recorded in the spectral range of 3900 cm⁻¹ to 800 cm⁻¹ at a spatial resolution of 11 µm and a spectral resolution of 6 cm⁻¹ using 3.5× (15× for high detail images; 5.5 µm digital resolution with 2 × 2 pixels binning) infrared magnification objectives (Bruker Optics). Each spectrum comprised 64 coadded scans. A reference of a single focal plane array window of the empty light path was acquired before image acquisition and automatically subtracted from the recorded image by the software OPUS (Bruker Optics). Atmospheric absorptions of water vapor and $CO_2$ were corrected by OPUS during image acquisition. OPUS files were imported into MatLab (The MathWorks) as ENVI files using the multiband-read function or by the irootlab toolbox[36]. Spectral features like carbohydrates and sucrose fingerprints along with baseline features were extracted using an extended multiplicative signal correction model[37] adopted into an in-house developed analytical MatLab routine for statistical and quantitative spectral feature analysis as described in ref. [23].

**Magnetic resonance imaging**. The non-invasive grain analysis relied on an Avance III HD 400 MHz NMR-spectrometer (Bruker). Structural ¹H imaging of the barley grains was performed using a multi-slice spin-echo sequence. The TR of the experiment was 0.5 s, TE = 7.4 ms, FOV = 15 × 9.5 × 9.5 mm, and 250 × 158 × 158 spatial points were acquired resulting in an in-plane resolution of 60 × 60 × 60 µm. Number of slices = 158, NA = 1, $T_{tot}$ = 6 h 56 min 4 s.

Dynamic ¹³C/¹H imaging experiments were performed by the detection of the ¹³C directly using an adjusted chemical shift imaging (CSI) protocol applied in spin-echo mode as described[1] in combination with cryogenically cooled double-resonant ¹H/¹³C probe (Bruker). As a reference image, a standard spin-echo sequence (MSME) was used with the following parameters: repetition time TR = 2500 ms, echo-time TE = 11 ms, matrix size 350 × 350, in-plane resolution 20 µm isotropic, slice thickness 300 µm. The SNR was improved by nuclear overhauser effect, a MLEV-pulse scheme was applied for 2000 ms on the ¹H channel. A 4 mm slice was excited with "calculated" pulses [excitation: pulse duration (tp) = 280 µs, bandwidth (BW) = 15 kHz; refocusing: tp = 227 µs, BW = 15 kHz). During acquisition, the MLEV scheme was also used for broadband ¹H signal decoupling. The TR of the sequence was 2.5 s, TE was 2.1 ms, number of averages 16. In all, 256 spectral points were acquired at a receiver bandwidth of 50 kHz. The experiments were performed with an isotropic FOV of 7 mm using an acquisition-weighted k-space sampling scheme. The in-plane resolution was 538 × 538 µm and the duration of the experiment 33 min 40 s. Data processing of NMR experiments was performed using MatLab (MathWorks). Segmentation and grain modeling was performed using AMIRA software (FEI Visualization Sciences Group).

**Measurement of grain ¹⁵N-Gln uptake**. For monitoring amino-acid uptake of the developing grains, stems were cut ~8 cm below the ear and placed in 5 mL of nutrient solution containing 20 mM $KH_2PO_4$, 100 mM sucrose, 10 mM MES buffer, 1 mM $CaCl_2$, pH 5.5, and 10 mM ¹⁵N-labeled Gln. During this period, the ears were held in the greenhouse. After 24 h, caryopses were isolated and separated into pericarp and endosperm fractions, lyophilized, and ground. The amount of ¹⁴N/¹⁵N in powdered samples was analyzed on an elemental analyser coupled to a stable isotope ratio mass spectrometer (Vario EL cube/Isoprime Vision, Elementar Analysensysteme). Fifteen biological repetitions with two technical replicates each were analyzed.

**NIRS**. Grain composition was measured in diffuse reflectance using a near-infrared spectroscope (Bruker MPA) in powdered samples according to the supplier's protocol (B-FING-M, Bruker) and using OPUS measurement software with integrated calibration.

**Metabolome analysis**. For the untargeted analysis of central metabolites, the freeze-dried samples were extracted as described[38], followed by ion chromatography (IC) using Dionex ICS-5000+ HPIC-system (Thermo Scientific) coupled to a QExactive-Plus hybrid quadrupole-orbitrap mass spectrometer (Thermo Scientific). The detailed chromatographic and MS conditions are described in Supplementary Table 3. The samples were randomized and analyzed in full MS mode. The data-dependent MS–MS analysis for the compound identification was performed in the pooled probe. The batch data were processed using the untargeted metabolomics workflow of the Compound Discoverer 3.0 software (Thermo Scientific). The alignment of the retention times was performed using an adaptive curve model with maximum shift of 0.5 min and 5 ppm mass tolerance. Feature detection was set at minimum peak intensity of 10,000, signal-to-noise threshold of 3 and 3 ppm mass tolerance. The peak area drift correction was performed using the cubic spline algorithm based on the repeated pooled probe as a quality control (QC). The compounds with the maximum relative standard deviation of 35% of the QC area were selected for the quantification. In addition, the overall stability of the sample recovery was monitored using the levulinic acid as an internal standard. The compounds were identified using an in-house spectral library and a mass list, as well as a public spectral database mzCloud and mass databases ChemSpider, KEGG, and Metabolika. The $P$ values of the group ratio were calculated by ANOVA and a Tukey-HCD post hoc analysis. In some cases, compounds were manually re-integrated using the TraceFinder4.1 software (Thermo Scientific) followed by the peak area correction using MetaboDrift Excel-software component[39]. Untargeted profiling of amino acids and some other cationic metabolites was performed using the Vanquish Focused ultra-high-pressure liquid chromatography system (Thermo Scientific) coupled to the same mass spectrometer. The batch processing and compound identification workflow were essentially the same as described for the IC-MS-based untargeted profiling.

**Phylogenetic analysis**. Predicted protein sequences were aligned using the ClustalX[40]. Accession numbers of used sequences are listed in Supplementary Table 4. The phylogenetic trees were constructed using the maximum-likelihood methods in MEGA6 with the following option settings: Poisson substitution model, uniform rates, partial deletion for gaps/missing data, 95% site coverage cutoff, strong branch swamp filter, and 1000 bootstrap replications.

**Statistics and reproducibility**. Statistical analysis and the number of technical and biological replicates for each experiment are indicated in figure legends and corresponding descriptions of methods. Phenotypic parameters of RNAi-VPE2-repressed grains were verified by analyzing independent batches of plants growing at different time periods. Besides four lines, analyzed in the manuscript, additional homozygous lines were grown and phenotypic parameters of their grains were evaluated. RNAi-mediated repression of *VPE2a*, *VPE2b*, and *VPE2d* genes was verified by qRT-PCR analysis of the grains from the independent batch of plants. All replicates were successful.

**Reporting summary**. Further information on research design is available in the Nature Research Reporting Summary linked to this article.

## Data availability

The authors declare that all data supporting the findings of the study are available within the paper and its Supplementary Information Files or from the corresponding authors on reasonable request. Source data for the main and supplementary figures are available in Supplementary Data 2.

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

## Acknowledgements

We are grateful to C. Matthess, S. Hermann, Carola Bollmann, and S. Tula for their excellent technical assistance, and A. Fiebig for help with the extraction of VPE sequences. This research was supported in part by Deutsche Forschungsgemeinschaft (BO1917/5-1 and BO1917/5-2).

## Author contributions

V.R. and L.B. conceived and designed the experiments. V.R., V.T., A.H., A.M., A.G., S.W., J.F., G.H., S.O., E.M., and H.R. performed the experiments. V.R., A.G., E.M., H.R., and L.B. analyzed the data. V.R., H.R., and L.B. wrote the article.

## Funding

## Competing interests

The authors declare no competing interests.
