## [Peer Review File · Communications Biology]

Reviewers' Comments:

Reviewer #1:

Remarks to the Author:

The Manuscript: "Grain filling in cereals relies on developmentally controlled programmed cell death" deals with the mechanism of delivery of assimilate to the endosperm. The experiments were designed to explore the importance of the VPE2a-VPE2d genes with respect to both the PCD of the NP and the removal of the cellular barriers obstructing the flow of assimilate from the maternal plant to the developing endosperm.

In general, the Manuscript is very well written and uses several different methods to support its hypothesis. All results are based on RNAi lines. I believe the manuscript results would be more convincing if overexpression, CRISPR knockout mutant or cosuppression lines could be compared in some of the experiments.

Major:

1) The description of PCD process in the NP cells is based only on electron micrographs Inspection. Cell organelles, such as cell membrane, ER, tonoplast, and nuclei, should be specifically labeled to support the conclusion of cellular disintegration is consistent with vacuole-mediated PCD.

2) Fig. 2: It is not clear why other VPE genes expression (VPE1, VPE3, VPE4, and VPE5) were analyzed in the barley plant's organs.

3) Lines 105-108: "There was no evidence of any phenotypic effect of the VPE2 subfamily down-regulation prior to anthesis, but once the grain filling phase began, the transgenic grains became distinct from the WT ones, appearing both thinner and longer." Eventually, the effect on gain weight is not very dramatic; this should be discussed

4) Fig. 4: Differences in maternal tissues and endosperm cell numbers looks minor. I believe it is hard to determine whether it is related to the assumption that "...fewer cells had been exposed to PCD". Maybe adding cumulative cell death/ PCD data to the same experiment will be more convincing.

5) Fig. 5b: most of the differences in the expression of transporter genes looks minor. No legends is included for Fig. 5b columns.

6) Fig. 7c needs more details in the legend; how many transgenic lines were analyzed? Only VPE2i-11?

Minor:

a) In general, i believe its better that all graphs containing 3 time points (DAF), in their X-axis, should be presented as a continuous line. It will better show the general pattern of change over DAF.

b) Line 41: "On the filial side of the crease, the vascular bundle is connected to a complex multicellular structure called the nucellar projection (NP) (Fig. 1a, Supplementary Movie 1)." NP is not mentioned in Movie 1.

c) Fig. 2: The full name of organism names (Os, Zm, Hv, Ta, Bd, etc.) should be written in the figure legend.

Reviewer #2:

Remarks to the Author:

The manuscript by Radchuk et al. analyzes programmed cell death in the nucellus projection of barley grains and describes its role in nutrient transport to the endosperm. The authors created knock-down lines for three genes encoding vacuolar processing enzymes that fail to undergo programmed cell death in the nucellus projection. Metabolic analyses of such transgenic lines revealed that the death of the nucellus projection affects nutrient transport and accumulation in the barley grain.

This is a novel, relevant and well conducted scientific work. The manuscript is well written and

easy to read. I am therefore favorable to publication after the following revisions:

MAJOR:

1) The authors should discriminate between cell elimination and PCD. In other words, death with or without the elimination of the cell wall. The barley NP appears to undergo both processes: cell elimination at the boundary with the endosperm and PCD in the rest of the tissue.

Two reviews on the subject:

-J Exp Bot. 2017 Feb 1;68(4):785-796.doi: 10.1093/jxb/erw364.

-Plant Reprod. 2018 Sep;31(3):309-317.doi: 10.1007/s00497-018-0338-1. Epub 2018 Jun 5.

2) Are VPE1, 3, 4, and 5 genes expressed in seeds? Is their expression affected in VPE2-RNAi lines?

3) The authors analyze different sets of transgenic lines in different experiments.

- Fig3A: all four lines

- Fig3B: only lines 11, 16, and 19

- Fig3E: only lines 11 and 19

- Fig4C: only lines 5 and 11

I understand that testing all four lines in every experiment is demanding, but I would have preferred to see analyzed always the same two or three lines. What is the authors' choice based on?

4) In line 124 the authors talk about "cell remnants" but they look like cells to my eyes.

Furthermore, Fig4A and SupFig2 show some empty spaces in the NP of VPE2i-11 seeds. The authors should address this point.

5) In figure 4B (lower picture, VPE2i-11) it is not easy to see the boundary between ETC and NP (especially considering the NP empty spaces mentioned above). Therefore, I find the authors' conclusions on the ETC morphology (line 139) not convincing.

6) Figures 6B and 6D show contradicting results on starch accumulation.

7) I encourage the authors to discuss more why, according to them, PCD and cell elimination in the nucellar projection improve nutrient transport to the fertilization products. Furthermore, the manuscript would benefit from a larger discussion at the light of what is known also in dicotyledonous seeds.

MINOR:

1) The authors should restrict their conclusions to barley and avoid generalizations, such as in the title: "cereals".

2) In line 63, the authors talk about the NP distal region. It looks to me that they refer to the medial-lateral polarity axis and not the proximal-distal one. I would include the axes in figure 1 to help the reader.

3) Authors should show bootstrap values at the nodes of their phylogenetic tree.

4) Figure 5B misses the transgenic lines color legend

5) The legends of figure 6B and 6C are swapped.

Reviewer #3:

Remarks to the Author:

This manuscript by Radchuk et al. describes the influence of programmed cell death on assimilate distribution from maternal tissues to seed storage tissue during seed development. They observed that cellular disintegration of the nucellar projection (NP) is driven by vacuolar-mediated programmed cell death (PCD). They previously identified a VPE2 subfamily in the Triticeae that is expressed in the NP. Here, they report that RNAi silencing of this VPE2 subfamily reduced grain weight and altered tissue composition of the grain. Notably, the RNAi lines also had aberrant PCD in the NP and a defective assimilate transport route - with inhibited sucrose transport to the endosperm and a concomitant reduction in starch accumulation. Generally, wide-range adjustments of the grain central metabolism were observed after VPE2 suppression. The authors conclude that PCD serves to widen the post-vascular assimilate route, accelerating assimilate allocation by dismantling the NP and removing a barrier on assimilate transport in front of the ETC. They claim that manipulating the timing of this PCD could represent a novel means of increasing the efficiency of the grain filling process in the major cereal crop species.

My overall impression is that this is an interesting and well-conducted study. I have some suggestions for the author to consider:

1. Line 99 says that the RNAi-fragment was from VPE2a, but the authors observed effective silencing of all VPE2 genes (Fig 3a). It is not clear from the text whether the RNAi fragment was designed intentionally so that all VPE2 isoforms could be suppressed, or whether the suppression of other VPE2s was accidental? However, given this observation, what is the guarantee that other isoforms such as HvVPE1 have also not been suppressed? Could this be determined by qPCR?
2. The phylogenetic tree has an irregularity where HvVPE2a and HvVPE2b cluster together. Thus, HvVPE2b ends up in the VPE2a clade rather than with the other Triticeae VPE2bs. It is therefore incorrect to simply shade the HvVPE2b into the VPE2b clade as the authors have done, because that is clearly not a clade supported by phylogeny. The authors should include bootstrap values in their tree - if there was strong bootstrap support that HvVPE2a clusters with HvVPE2b, then the authors must change their interpretation and conclude that the two isoforms result from a much more recent duplication in barley. However, if there was poor bootstrap support (which I suspect), then it is simply the case that HvVPE2b has ended up in the wrong place in the tree.
3. In Figure 5a, it is unclear to me why no labelled sucrose is detected at all in the 1 h timepoint in the RNAi line. Would the authors not expect to at least see it in the rachis? Or is transport up the rachis also affected in the RNAi line?
4. It is a strength of this paper that most analyses were carried out on multiple independent RNAi lines. Understandably, the more technically challenging analyses (e.g., Fig 3c, 3d, 4a, 4b, 5a), were conducted on one of the lines, mostly VPE2i-11. It would be good if the authors clarify in the text why this line in particular was chosen.
5. Line 117 says no difference in storage components in transgenic lines compared to WT. But, supplementary fig. 1 shows statistically significant differences in starch, protein and lipid, except for line number 5? If this is reason why line 5 was excluded in most other experiments, this could be mentioned?
6. Authors should check their wording to ensure that they do not exaggerate their findings. For example: Line 96, "only detectable in the developing caryopses" directly contradicts the previous statement that all genes were expressed in the anthers. Line 126 suggests that disintegrating nuclei are not present in transgenic lines, but Supplemental Fig 2 legend says they are "almost absent". Line 143: "VPE activity is essential for PCD" – essential is too strong as it suggests absolute requirement, but some PCD still happens after VPE2 is suppressed.
7. Regarding Figure 6c, I find this a major distraction in an otherwise very convincing figure. Surely the most likely explanation for the reduced starch synthesis in the RNAi lines is the

defective sucrose transport that the authors convincingly present in Fig 5 and Fig 6 d? The extent to which those levels of AGPase transcript reduction in Fig 6C can really explain to the observed changes on starch content is not known. Also, the rationale for measuring GBSS – a gene that has no impact on total starch content – is also unclear (Also note that the pericarp-specific GBSS is more often referred to as GBSS2 rather than GBSS1b). I feel the authors can afford to remove this part without losing any meaning or support for their main conclusions.

8. Related to point 7, I personally did not feel the metabolomics in Fig 7 adds too much to the story, especially since this large dataset is not extensively discussed in the paper, and most changes cannot be clearly explained. Of course, this is completely up to the authors whether they want to include it here, or perhaps save it for a follow-up paper.

9. Materials and Methods does not have a description of Caspase assays.

Minor points:

Regarding italicisation: Throughout the manuscript, the authors are not consistent with their italicisation rules. Check with journal standards, but in general, species/genus names should always be italicised, but families not. Genes should be, but not proteins. Examples include Line 88, where VPE2 subfamily is not in italics even though it was italicised in line 85; Line 89, where Triticeae is once written in italics, then once written without; Supplemental Fig 1, Supplemental Fig 3 legend

Line 42: Supplementary movie 1 is cited here, but crease and NP not obvious in the movie. Can these be labelled?

Fig 1: Staining method for panel a should be mentioned in the legend (Line 519). Legend should also specify what the red circle in panel e indicates? Surely that is not a VB as in panel a?

Figure legends: Generally, the inclusion of the developmental stage of grains (in DAF) in relevant figure legends would be useful.

Line 67: presaging is an uncommon word.

Line 87: Aegilops is abbreviated with Ae. instead of A.; which is fine, but then in the phylogenetic tree, it should be AesVPE rather than AsVPE?

Fig 2: There is a hidden white "V" visible in the yellow shading around the VPE2a clade and VPE2d clade. In the tree, why is there a number 4 added after names of TaVPE1A and TaVPE1D? Panel B has expression in relative signal intensity [%] while other qPCR figures show relative transcript abundance - if possible make uniform. Statistical analyses are missing – would be useful for comparing the different VPE2 isoforms within each tissue.

Regarding supplementary tables: Check naming and text references – currently there are two tables named supplementary table 3. In the supposed Supp Table 4, why do some gene IDs have splice variants indicated, and some not?

Fig 3 e) graph for pericarp is slightly cut off – the 0 is clipped

Supplementary Fig 1: b) this graph is not mentioned in text

Fig 4 legend: missing abbreviation for ec 'endosperm cavity'. Abbreviations NP, ETC in legend are capitalised, but not in figure and abbreviations list? WT not in abbreviations list.

Line 135: "both genes were clearly less strongly transcribed in the transgenic grains" – clarify here that this is for most timepoints. Similar for the statements on Lines 142 and 186.

Fig 5a: would be easier to understand if timepoints were labelled along the vertical axis.

Fig 5b: The legend that specifies which bars represent which RNAi-lines is missing

Line 164: check apostrophes on latter

Line 171: There appears to be a mixup between Fig 6b and 6c? Check this throughout manuscript. Also, these two are the wrong way around in the figure legend.

Fig 6d: tissues should be labelled.

Line 183: "While the global uptake of Gln was largely unaffected" – unclear which data this is referring to.

Figs 7 and S3: be consistent with capitalisation of hordein between figure and legend. In S3 legend, abbreviation should be CAT1 rather than CAT.

Line 186: add abbreviations of gene names in brackets

line 254 : omit "of"

line 332 and 334: replace squares with hyphens

Reviewer #1 (Remarks to the Author):

The Manuscript: "Grain filling in cereals relies on developmentally controlled programmed cell death" deals with the mechanism of delivery of assimilate to the endosperm. The experiments were designed to explore the importance of the VPE2a-VPE2d genes with respect to both the PCD of the NP and the removal of the cellular barriers obstructing the flow of assimilate from the maternal plant to the developing endosperm.

In general, the Manuscript is very well written and uses several different methods to support its hypothesis. All results are based on RNAi lines. I believe the manuscript results would be more convincing if overexpression, CRISPR knockout mutant or cosuppression lines could be compared in some of the experiments.

Response: The genes encoding VPE2a, VPE2b and VPR2d have very similar temporal and spatial expression patterns, which suggests some highly redundant function in the nucellar projection of the barley grain. We therefore aimed to modulate all three genes at once. The knock-down of all three genes at once was possible by the well-established RNAi approach, but CRISPR technology was not yet available for that purpose in barley at that time. An advantage of the RNAi approach is also that the knock-down generally leads to a less severe phenotype as compared to a knock-out (which is often lethal and thus not desired). An overexpression approach is underway, but this will take at least two years before we will have respective lines available.

Major:

1) The description of PCD process in the NP cells is based only on electron micrographs Inspection. Cell organelles, such as cell membrane, ER, tonoplast, and nuclei, should be specifically labeled to support the conclusion of cellular disintegration is consistent with vacuole-mediated PCD.

Response: The reviewer is correct, that there are distinct labelling procedures available for the various cell organelles. Each of these is a challenge for application on non-model plant systems. We have decided here to trace ultrastructural features of cells/organelles, and the best and safest method for this is electron microscopy. All our EM-related findings were compared to the classification system provided in (1) van Doorn et al. Morphological classification of plant cell deaths. Cell Death Differ. 2011; and (2) Kabbage et al. The life and death of a plant cell. Annu. Rev. Plant Biol. 2017. In addition, we have identified vacuolar enzymes as key players in the PCD process. Finally, we have to point out that the aim of this study was to identify the role of PCD (and VPE in particular) for grain filling. Thus, we did not intend to stress the mode of PCD as this is of secondary importance for us. If the reviewer insists to remove the aspect of "vacuole-mediated", we are ready to remove this.

We have added subfigures to the Fig. 1 to document the sequence of PCD in more detail. We have added labelling of cell organelles at this figure. We also have prepared a new Supplementary Fig. 1, which additionally illustrates the sequence of PCD events in the nucellar projection.

2) Fig. 2: It is not clear why other VPE genes expression (VPE1, VPE3, VPE4, and VPE5) were analyzed in the barley plant's organs.

Response: We aimed to provide full, comprehensive view on the VPE2 gene subfamily in Fig. 2. To demonstrate their expansion, we had to include all other VPE genes in the bootstrap consensus tree of the VPE protein sequences from barley and related species (Fig. 2a). The expression data were only given for the VPE2 subfamily (Fig. 2b). The other VPE-encoding genes are only given in the new Supplementary Fig. 2 – this demonstrates that VPE1, VPE3, VPE4, and VPE5 are abundantly expressed in a number of vegetative and regenerative tissues but nearly absent in the nucellar projection.

3) Lines 105-108: "There was no evidence of any phenotypic effect of the VPE2 subfamily down-regulation prior to anthesis, but once the grain filling phase began, the transgenic grains became distinct from the WT ones, appearing both thinner and longer." Eventually, the effect on grain weight is not very dramatic; this should be discussed

Response: We added corresponding discussion to this aspect (page 11). There are two reasons why the effect on grain weight is not very dramatic: (1) we have achieved moderate suppression of VPE2 subfamily expression (Fig. 3) but not a complete knock-out. (2) The expression of relevant transporters (SWEET11a and SWEET11b) was increased in the VPE2-affected grains (Fig. 5b). These sugar exporters are expressed in the nucellar projection (Mascher et al., 2017), and likely compensate the negative effect of VPE2-repression on sucrose delivery.

4) Fig. 4: Differences in maternal tissues and endosperm cell numbers looks minor. I believe it is hard to determine whether it is related to the assumption that "...fewer cells had been exposed to PCD". Maybe adding cumulative cell death/ PCD data to the same experiment will be more convincing.

Response: Indeed, the increase in cell number especially in maternal tissues appears minor. But we collected additional evidence for an interesting compensatory effect occurring in the transgenics: total cell number in maternal tissue (Fig.4C) is mainly resulting from nucellar projection and pericarp tissues. Cell number in nucellar projection is elevated due to VPE2 repression. However, this is counterbalanced by reduced cell number in pericarp tissue. The latter is a consequence of upregulated expression of VPE4 (new dataset in Suppl. Fig. 5). VPE4 is predominantly expressed in the pericarp and known to eliminate cells via PCD in that tissue (for details Radchuk et al., New Phytologist 2018). These two combined effects can explain the rather mild effect on total cell numbers in transgenics. We discuss this on page 7.

5) Fig. 5b: most of the differences in the expression of transporter genes looks minor. No legends is included for Fig. 5b columns.

Response: Fig. 5b actually shows that differences in the expression of sugar-transport-related genes are - depending on stage – rather strong: almost 4-fold for SUT1, and more ~ 3-fold for both SWEET11a and SWEET 11b. These changes were statistically highly significant. Based on our experience on sugar transporters in developing seeds such changes can substantially affect grain filling/development, and thus are not minor.

Of course, the reasons for these changes in expression of sugar transporters are quite intriguing. PCD affects the distribution of sugars within the grain tissues. Local concentrations in turn could affect the expression of respective genes. The transgenic lines could therefore represent an interesting model to further investigate such aspects of gene expression regulation in an important cereal crop.

We have added the legend to the Fig. 5b.

6) Fig. 7c needs more details in the legend; how many transgenic lines were analyzed? Only VPE2i-11?

Response: Fig. 7c indeed summarizes metabolite data on line VPE2i-11 in comparison to WT. We have added this information to the legend.

Minor:

a) In general, I believe it's better that all graphs containing 3 time points (DAF), in their X-axis, should be presented as a continuous line. It will better show the general pattern of change over DAF.

Response: we improved all graphs according to the style rules of the journal. When figures are presented with a continuous line, this led to overload (all single data points, SD and statistic). We therefore decided to keep the representation (with bars).

b) Line 41: "On the filial side of the crease, the vascular bundle is connected to a complex multicellular structure called the nucellar projection (NP) (Fig. 1a, Supplementary Movie 1)." NP is not mentioned in Movie 1.

Response: Thanks for this, we follow your recommendation and improved this movie by (1) mentioning the NP, and (2) including a reference image, showing an enlarged view of the crease region of barley grain.

c) Fig. 2: The full name of organism names (Os, Zm, Hv, Ta, Bd, etc.) should be written in the figure legend.

Response: We added the full organism names to the figure legend as recommended.

Reviewer #2 (Remarks to the Author):

The manuscript by Radchuk et al. analyzes programmed cell death in the nucellus projection of barley grains and describes its role in nutrient transport to the endosperm. The authors created knock-down lines for three genes encoding vacuolar processing enzymes that fail to undergo programmed cell death in the nucellus projection. Metabolic analyses of such transgenic lines revealed that the death of the nucellus projection affects nutrient transport and accumulation in the barley grain.

This is a novel, relevant and well conducted scientific work. The manuscript is well written and easy to read. I am therefore favorable to publication after the following revisions:

MAJOR:

1) The authors should discriminate between cell elimination and PCD. In other words, death with or without the elimination of the cell wall. The barley NP appears to undergo both processes: cell elimination at the boundary with the endosperm and PCD in the rest of the tissue.

Two reviews on the subject:

-J Exp Bot. 2017 Feb 1;68(4):785-796. doi: 10.1093/jxb/erw364.

-Plant Reprod. 2018 Sep;31(3):309-317. doi: 10.1007/s00497-018-0338-1. Epub 2018 Jun 5.

Response: We appreciate this advice very much, and included this aspect to the first paragraph of Results (page 4) and Discussion (p. 11). This helps to bring more clarity in the issue. In general, cell death events could be classified based on their outcome: cell elimination, during which cell corpses are completely degraded and the structure-conserving cell death. Here we observed how both processes (death with or without the elimination of the cell wall) congregate in a distinct temporal order in the NP of barley. Cells of nucellar projection undergo PCD and lose their integrity gradually in accordance to a developmental gradient. During this time, they maintain their structural role as a conduit for the allocation of nutrients unloaded from the vascular bundle. During development, most cells of nucellar projection are completely degraded (eliminated) at the boundary to the endosperm. Such a reduction of the post-phloem pathway implies facilitated nutrient allocation toward the main sink (endosperm) and promoted grain filling. VPE2-RNAi decelerates execution of PCD in the NP, and helped to display the biological relevance of the entire process.

2) Are VPE1, 3, 4, and 5 genes expressed in seeds? Is their expression affected in VPE2-RNAi lines?

Response: We have analysed the expression of the other VPE genes in vegetative tissues as well as in the developing grains including the VPE2-RNAi lines. We have put these data to supplementary Fig. 2. The outcome was as follows. The VPE1, 3, 4, and 5 genes were expressed in various vegetative tissues (Suppl. Fig. 2a). In the microdissected NP region, we see abundant expression of VPE2a, VPE2b and VPE2d, but negligible expression of other VPE genes (Suppl. Fig. 2b). This let us conclude that all observed changes are related to the repression of NP-specific VPE2a, VPE2b and VPE2d.

The new dataset also revealed some expression changes for VPE4 in our transgenics (Suppl. Fig. 5). VPE4 is expressed in the pericarp (not in NP) and known to eliminate cells via PCD in that tissue (for all details Radchuk et al., New Phytologist 2018). The increased expression of the pericarp-specific VPE4 in VPE2-repressed grains imply some accelerated PCD in the pericarp. This analysis has helped us to explain minor changes in the number of maternal cells of transgenic grains when compared with WT. We discuss this on page 7 (see also response #4 to Reviewer #1).

3) The authors analyze different sets of transgenic lines in different experiments.

- Fig3A: all four lines
- Fig3B: only lines 11, 16, and 19
- Fig3E: only lines 11 and 19
- Fig4C: only lines 5 and 11

I understand that testing all four lines in every experiment is demanding, but I would have preferred to see analyzed always the same two or three lines. What is the authors' choice based on?

Response: Most of the data are presented now for all 4 transgenic lines. The time/resource demanding in vivo MRI measurement experiments and FTIR chemical imaging was done for 2 transgenic lines + WT; MRI-based flow measurements, isotope feeding ($^{13}\text{C}/^{15}\text{N}$) and metabolomics was done for only one strongest line +WT. In detail:

- Fig3A: all four lines
- Fig3B: all four lines
- Fig3E: two lines (#11 and #19 both with strong phenotype)
- Fig3F: all four lines
- Fig.4B: two lines (#11 and #19 both with strong phenotype)
- Fig4C: two lines (#11 and # 5; these are representative for a strong phenotype (#11) and a weak phenotype (#5))
- Fig4D: all four lines
- Fig4E: all four lines
- Fig5A: only one line (#11) was used for the time/resource demanding MRI experiments
- Fig.5B: all four lines
- Fig.6A: only one line (#11) was used for the time/resource demanding metabolomics
- Fig.6B: all four lines
- Fig.6C: all four lines
- Fig.6D: only one line (#11) was used for FTIR-based imaging
- Fig.7A: only one line (#11) was used for isotope tracer experiments
- Fig.7B: all four lines
- Fig.7A: only one line (#11) was used for metabolomics
- Suppl. Fig.2: two lines (#11 and #19 for TUNEL assay)
- Suppl. Fig.5: all four lines

In summary, with the additional data included now, we can present a more comparable dataset (with data on identical lines).

4) In line 124 the authors talk about "cell remnants" but they look like cells to my eyes. Furthermore, Fig4A and SupFig2 show some empty spaces in the NP of VPE2i-11 seeds. The authors should address this point.

Response: We agree with the reviewer and changed wording into: “The consequence was that non-eliminated cells accumulated in the apoplastic cavity of the transgenic grains”. The image in Fig. 4A indeed allows to see the non-digested cells in transgenics. In contrast, cell elimination in WT occurred more efficiently and endosperm cavity fluid was mostly cleared of cell debris.

Regarding the “empty spaces”: Impaired PCD could easily affects shape and mechanical properties of the NP tissues. In transgenic plants, where cell processing/PCD is affected, the structural arrangement and properties are also altered. The empty spaces in the NP of VPE2i-11 seeds are likely due to the irregular/abberant and fragile tissue structure of the NP in transgenics.

5) In figure 4B (lower picture, VPE2i-11) it is not easy to see the boundary between ETC and NP (especially considering the NP empty spaces mentioned above). Therefore, I find the authors’ conclusions on the ETC morphology (line 139) not convincing.

Response: We accept the criticism of the reviewer. Our conclusion is based on observation on three transgenic lines (VPE2i-11, VPE2i-19, VPE2i-5 vs WT). Our statement cited above is best and clearly seen in VPE2i-11 and VPE2i-19 lines. We have added the histological sections for the VPE2i-19 line (and corresponding WT) (new Fig.4B lower panels and new Supplementary Fig. 6), which in particular highlight the cellular structure of the ETC.

6) Figures 6B and 6D show contradicting results on starch accumulation.

Response: To avoid misunderstanding we have provided additional information in Fig. 6D. To explain: the Fig.6b shows the starch content in whole grains, indicating a clearly lower starch level in the developing caryopsis. To check specifically starch levels in both pericarp and endosperm tissues, FTIR-based imaging of starch distribution was done. The quantification of starch in these two tissue types across 5 biological replicates is represented in the new Fig. 6D – it indicates a statistically significantly higher starch content in pericarp but less starch in endosperm of transgenic vs WT grains. The exemplary images in Fig. 6E show how the gradients look like in the region of interest. As the endosperm makes a bigger biomass contribution than the pericarp, the starch content of the entire caryopsis is lower in transgenics (as seen on Fig. 6B).

7) I encourage the authors to discuss more why, according to them, PCD and cell elimination in the nucellar projection improve nutrient transport to the fertilization products. Furthermore, the manuscript would benefit from a larger discussion at the light of what is known also in dicotyledonous seeds.

Response: PCD and cell elimination in the nucellar projection improve nutrient transport to the fertilization products because cells of NP are disintegrated, making the entire NP more permeable to assimilate flow/transport. Thereby, a major (maternal) barrier to the transport of assimilate is removed, allowing the endosperm to drive the process of grain filling. We further added some discussion to that subject (page 11). Space constraints prevent us from providing a larger discussion of what is known about dicotyledonous seeds, but we refer to a recent review paper to guide the reader and assess the complexity of issues.

MINOR:

1) The authors should restrict their conclusions to barley and avoid generalizations, such as in the title: “cereals”.

Response: We have changed the title. Wording in conclusions is improved to avoid unnecessary generalizations.

2) In line 63, the authors talk about the NP distal region. It looks to me that they refer to the medial-

lateral polarity axis and not the proximal-distal one. I would include the axes in figure 1 to help the reader.

Response: Thanks for recommendation, we have added the axes in Fig. 1a and 1b.

3) Authors should show bootstrap values at the nodes of their phylogenetic tree.

Response: We have now added bootstrap values to the Fig. 2a, as recommended.

4) Figure 5B misses the transgenic lines color legend

Response: We have added the missing colour legend to the Figure, as recommended.

5) The legends of figure 6B and 6C are swapped.

Response: We are sorry for this; we have corrected the figures.

Reviewer #3 (Remarks to the Author):

This manuscript by Radchuk et al. describes the influence of programmed cell death on assimilate distribution from maternal tissues to seed storage tissue during seed development. They observed that cellular disintegration of the nucellar projection (NP) is driven by vacuolar-mediated programmed cell death (PCD). They previously identified a VPE2 subfamily in the Triticeae that is expressed in the NP. Here, they report that RNAi silencing of this VPE2 subfamily reduced grain weight and altered tissue composition of the grain. Notably, the RNAi lines also had aberrant PCD in the NP and a defective assimilate transport route - with inhibited sucrose transport to the endosperm and a concomitant reduction in starch accumulation. Generally, wide-range adjustments of the grain central metabolism were observed after VPE2 suppression. The authors conclude that PCD serves to widen the post-vascular assimilate route, accelerating assimilate allocation by dismantling the NP and removing a barrier on assimilate transport in front of the ETC. They claim that manipulating the timing of this PCD could represent a novel means of increasing the efficiency of the grain filling process in the major cereal crop species.

My overall impression is that this is an interesting and well-conducted study. I have some suggestions for the author to consider:

1. Line 99 says that the RNAi-fragment was from VPE2a, but the authors observed effective silencing of all VPE2 genes (Fig 3a). It is not clear from the text whether the RNAi fragment was designed intentionally so that all VPE2 isoforms could be suppressed, or whether the suppression of other VPE2s was accidental? However, given this observation, what is the guarantee that other isoforms such as HvVPE1 have also not been suppressed? Could this be determined by qPCR?

Response: Knowing overlapping patterns of VPE2a-VPE2d expression, we have intentionally designed the construct in such way that allows repression of the whole subfamily. The selected RNAi fragment was ~93-97% identical to the corresponding regions of VPE2b and VPE2d genes. We have mentioned this in the manuscript. The selected VPE2a fragments cannot target any of other VPE genes because of low similarity among their regions (see also phylogenetic tree, Fig. 2a). Further, the selected VPE2a promoter is not active in other tissues, beside the NP, making it very unlikely to target (repress) any of the other VPE genes.

Regardless of these considerations, we followed the recommendation of the reviewer and performed some additional qPCRs (Supplementary Fig. 5). The data on transcript abundancies of the other VPE

genes revealed that expression of VPE1 and VPE3 was not affected by the transgenic approach. Moreover, the expression of VPE4 was even increased.

2. The phylogenetic tree has an irregularity where HvVPE2a and HvVPE2b cluster together. Thus, HvVPE2b ends up in the VPE2a clade rather than with the other Triticeae VPE2bs. It is therefore incorrect to simply shade the HvVPE2b into the VPE2b clade as the authors have done, because that is clearly not a clade supported by phylogeny. The authors should include bootstrap values in their tree - if there was strong bootstrap support that HvVPE2a clusters with HvVPE2b, then the authors must change their interpretation and conclude that the two isoforms result from a much more recent duplication in barley. However, if there was poor bootstrap support (which I suspect), then it is simply the case that HvVPE2b has ended up in the wrong place in the tree.

Response: We agree with the reviewer, and have removed the shading in Fig. 2A and included bootstrap values. The phylogenetic tree does not allow clear conclusion when and how VPE2a and VPE2b were separated in relation to their orthologs/homologs from other Triticeae.

3. In Figure 5a, it is unclear to me why no labelled sucrose is detected at all in the 1 h timepoint in the RNAi line. Would the authors not expect to at least see it in the rachis? Or is transport up the rachis also affected in the RNAi line?

Response: The pattern, where 13C labelled sucrose is not accumulating in the rachis of WT and is detectable at time points later than 1h in transgenic grains, is robust. It could be explained by the following mechanistic model: the transgenic grain is already substantially loaded with sucrose (up to 3-fold more sucrose than in WT based on metabolite profiling data). This may generally lead to a lower withdrawal of sucrose from the supplying vascular bundle (rachis) because the driving force (i.e. concentration difference) for sucrose uptake into the caryopsis is lower. Consequently, the overall 13C-sucrose transport toward the transgenic caryopsis is lowered. When we not yet see a signal for 13C-labelled sucrose in the rachis at 1h, this means the 13C amount is still BELOW the detection threshold of MRI. Only at later time points, the signal in rachis has increased to a detectable level. This evidences successive accumulation of labelled sucrose in rachis while sucrose transfer toward and within the caryopsis is slower.

This scenario has an analogy from everyday experience (non-plant world): when a highway exit becomes (partially) blocked, vehicles start to "accumulate" near the exit. Eventually, a tailback of cars develops along the highway and traffic slows down (in analogy to the RNAi line: our 13C molecule delays due to the already existing "sucrose traffic jam" in the rachis).

4. It is a strength of this paper that most analyses were carried out on multiple independent RNAi lines. Understandably, the more technically challenging analyses (e.g., Fig 3c, 3d, 4a, 4b, 5a), were conducted on one of the lines, mostly VPE2i-11. It would be good if the authors clarify in the text why this line in particular was chosen.

Response: We have analysed more lines whenever possible. For time- and/or resource-demanding experiments we have analysed only line VPE2i-11, because this line showed the strongest phenotype. In some cases, the results achieved with this line were supported by analysis of another line with strong phenotype (VPE2i-19) or the weaker lines VPE2i-16 and VPE2i-5. We have now added all new data to the figures and to the supplementary data. The reasoning for choosing line VPE2i-11 is mentioned in the manuscript (page 5).

5. Line 117 says no difference in storage components in transgenic lines compared to WT. But, supplementary fig. 1 shows statistically significant differences in starch, protein and lipid, except for line number 5? If this is reason why line 5 was excluded in most other experiments, this could be mentioned?

Response: The former Supplementary Fig. 1 is now Supplementary Fig. 3, and it shows the accumulation of starch, protein and lipid on a per grain level. Due to the lower grain weight, we see here a reduction on storage compounds. However, the relative content of these compounds (on a per gram level) was not affected in the transgenics. We rephrased the respective sentences in the revised manuscript to avoid misunderstanding (page 6).

The line 5 generally showed a weak phenotype, but the tendency in lower accumulation of storage compounds was the same.

6. Authors should check their wording to ensure that they do not exaggerate their findings. For example: Line 96, “only detectable in the developing caryopses” directly contradicts the previous statement that all genes were expressed in the anthers. Line 126 suggests that disintegrating nuclei are not present in transgenic lines, but Supplemental Fig 2 legend says they are “almost absent”. Line 143: “VPE activity is essential for PCD” – essential is too strong as it suggests absolute requirement, but some PCD still happens after VPE2 is suppressed.

Response: We have changed wording to avoid overinterpretations.

7. Regarding Figure 6c, I find this a major distraction in an otherwise very convincing figure. Surely the most likely explanation for the reduced starch synthesis in the RNAi lines is the defective sucrose transport that the authors convincingly present in Fig 5 and Fig 6 d? The extent to which those levels of AGPase transcript reduction in Fig 6C can really explain to the observed changes on starch content is not known. Also, the rationale for measuring GBSS – a gene that has no impact on total starch content – is also unclear (Also note that the pericarp-specific GBSS is more often referred to as GBSS2 rather than GBSS1b). I feel the authors can afford to remove this part without losing any meaning or support for their main conclusions.

Response: Our data clearly evidences that starch accumulation is reduced in transgenics at the early developmental stages (3-12DAF; Fig. 6B). The look into transcript abundances of starch-related genes was therefore a logical step (Fig. 6C). Our data show that roughly 2-fold decrease in transcript abundance AGP-S1a (encoding a subunit of the rate-limiting enzyme) corresponds to a one-third decrease in starch accumulation. This indicates the extent to which transcript abundance relates to the function (activity) of the encoded enzyme.

The rationale to analyses the GBSS 1a and 1b was the following. The two isoforms are genes from starch biosynthesis machinery with clearly differential expression in the developing grain: GBSS1a is only active in the pericarp while GBSS1b is strongly endosperm-specific (Radchuk et al., 2009). The two isoforms GBSS1a and GBSS1b in barley were named many years ago (Patron et al., 2002); this nomenclature is broadly used throughout barley research, and we therefore we have left these names.

8. Related to point 7, I personally did not feel the metabolomics in Fig 7 adds too much to the story, especially since this large dataset is not extensively discussed in the paper, and most changes cannot be clearly explained. Of course, this is completely up to the authors whether they want to include it here, or perhaps save it for a follow-up paper.

Response: Due to space limitations we could not describe the metabolic dataset in more details, but we are truly convinced that this is of value and of particular interest to some researchers working on barley grain metabolism. We therefore would like to keep this part as it is.

9. Materials and Methods does not have a description of Caspase assays.

Response: Thank you very much for your sharp look, this was indeed somehow missing. Now we have added this information to the Material and Methods section.

Minor points:

Regarding italicisation: Throughout the manuscript, the authors are not consistent with their italicisation rules. Check with journal standards, but in general, species/genus names should always be italicised, but families not. Genes should be, but not proteins. Examples include Line 88, where VPE2 subfamily is not in italics even though it was italicised in line 85; Line 89, where Triticeae is once written in italics, then once written without; Supplemental Fig 1, Supplemental Fig 3 legend

Response: we have rechecked the italicisation and corrected the mistakes. We have written “VPE2 family” not italic because this is a name of the subfamily but not a gene.

Line 42: Supplementary movie 1 is cited here, but crease and NP not obvious in the movie. Can these be labelled?

Response: we have added labelling of the NP and also included a reference image into the movie.

Fig 1: Staining method for panel a should be mentioned in the legend (Line 519). Legend should also specify what the red circle in panel e indicates? Surely that is not a VB as in panel a?

Response: the staining method is added to the legend. The red circle was removed. The entire figure was rearranged/improved.

Figure legends: Generally, the inclusion of the developmental stage of grains (in DAF) in relevant figure legends would be useful.

Response: we have added developmental stages (in DAF).

Line 67: presaging is an uncommon word.

Response: We do not have a better word, but would appreciate any suggestion from the reviewer.

Line 87: Aegilops is abbreviated with Ae. instead of A.; which is fine, but then in the phylogenetic tree, it should be AesVPE rather than AsVPE?

Response: we have corrected AsVPE to AesVPE in the Fig. 2a.

Fig 2: There is a hidden white “V” visible in the yellow shading around the VPE2a clade and VPE2d clade. In the tree, why is there a number 4 added after names of TaVPE1A and TaVPE1D? Panel B has expression in relative signal intensity [%] while other qPCR figures show relative transcript abundance - if possible make uniform. Statistical analyses are missing – would be useful for comparing the different VPE2 isoforms within each tissue.

Response: We have remade this Fig. 2a and corrected all mistakes. We also recalculated the data in Fig. 2b to represent qPCR results in relative values. As we have no WT/transgenics comparison here and the expression of the VPE genes is independent from each other, we have not added statistical analysis.

Regarding supplementary tables: Check naming and text references – currently there are two tables named supplementary table 3. In the supposed Supp Table 4, why do some gene IDs have splice variants indicated, and some not?

Response: We have rechecked all supplementary data and references for it throughout the text. Please note that we have added several new supplementary data, so that numbers have shifted.

Fig 3 e) graph for pericarp is slightly cut off – the 0 is clipped

Response: Corrected.

Supplementary Fig 1: b) this graph is not mentioned in text

Response: This is now mentioned in the text, first paragraph of Results.

Fig 4 legend: missing abbreviation for ec 'endosperm cavity'. Abbreviations NP, ETC in legend are capitalised, but not in figure and abbreviations list? WT not in abbreviations list.

Response: Corrected.

Line 135: “both genes were clearly less strongly transcribed in the transgenic grains” – clarify here that this is for most timepoints. Similar for the statements on Lines 142 and 186.

Response: Corrected.

Fig 5a: would be easier to understand if timepoints were labelled along the vertical axis.

Response: We added this information to the legend already.

Fig 5b: The legend that specifies which bars represent which RNAi-lines is missing.

Response: Corrected.

Line 164: check apostrophes on latter

Response: Corrected.

Line 171: There appears to be a mixup between Fig 6b and 6c? Check this throughout manuscript. Also, these two are the wrong way around in the figure legend.

Response: Corrected.

Fig 6d: tissues should be labelled.

Response: Description of the tissues is added.

Line 183: “While the global uptake of Gln was largely unaffected” – unclear which data this is referring to.

Response: We rephrased this sentence for better understanding.

Figs 7 and S3: be consistent with capitalisation of hordein between figure and legend. In S3 legend, abbreviation should be CAT1 rather than CAT.

Response: Corrected.

Line 186: add abbreviations of gene names in brackets

Response: Corrected.

line 254 : omit “of”

Response: Corrected.

line 332 and 334: replace squares with hyphens

Response: Corrected.

REVIEWERS' COMMENTS:

Reviewer #2 (Remarks to the Author):

I thank the authors for addressing all the concerns I raised. I am favorable to the publication of this version of the manuscript.

Reviewer #3 (Remarks to the Author):

The authors convincingly revised and improved the manuscript in accordance to the reviewers' suggestions. I am happy to recommend this study for publication and would like to congratulate the authors on an excellent piece of work.

I would like to add a few comments for the final manuscript:

1. I would highly recommend omitting the sentence "It seems likely that VPE2a, VPE2b and VPE2d evolved as a result of two independent duplication events, with the creation of VPE2d predating that of VPE2a and VPE2b." In the revised tree, this statement is likely true for the barley VPE2s, but not the other cereals. The authors already mentioned in their response that the phylogenetic tree does not allow a clear conclusion when and how VPE2a and VPE2b were separated in relation to their orthologs/homologs from other Triticeae. I agree with their response, and thus feel the sentence in the manuscript is incorrect, and anyway can be removed without losing any meaning from the study.
2. No changes in the manuscript are necessary, but just to respond to the rebuttal to point 7: Given the defective sucrose transport in the RNAi lines, it is difficult to conclude a direct correlation between decreased AGPase transcript abundance and decreased starch accumulation. The simplest explanation is that the reported decrease in starch content is related to the decreased sucrose transport. However, if the decrease in AGPase transcript abundance corresponds to the decrease in starch accumulation, the model becomes more complicated: Does this mean that the sucrose regulates AGPase at the transcriptional level (and the reduction in AGPase transcript is causative of the reduced starch) OR is the AGPase transcript level decreased due to some sort of feedback regulation from reduced starch accumulation (in which case the reduced sucrose supply is causative of the reduced starch content, and the AGPase transcript is a consequence). In their response, the authors seem to imply that it is the former - that reduced AGPase transcript that can explain the observed decrease in starch content. But in my opinion, there is not enough evidence to conclude one or the other.
3. I noticed in Figure 1 the abbreviation AC for endosperm cavity while in S1 it was apoplastic cavity. Are these expressions used synonymously?

Responses to Reviewers

We are grateful for very careful reading and constructive criticism of our manuscript, which has helped us to significantly improve the manuscript. Our answers on last comments are below.

Reviewer #3 (Remarks to the Author):

The authors convincingly revised and improved the manuscript in accordance to the reviewers' suggestions. I am happy to recommend this study for publication and would like to congratulate the authors on an excellent piece of work.

Response: We are happy to read the positive conclusion about our work.

Would like to add a few comments for the final manuscript:

1. I would highly recommend omitting the sentence "It seems likely that VPE2a, VPE2b and VPE2d evolved as a result of two independent duplication events, with the creation of VPE2d predating that of VPE2a and VPE2b." In the revised tree, this statement is likely true for the barley VPE2s, but not the other cereals. The authors already mentioned in their response that the phylogenetic tree does not allow a clear conclusion when and how VPE2a and VPE2b were separated in relation to their orthologs/homologs from other Triticeae. I agree with their response, and thus feel the sentence in the manuscript is incorrect, and anyway can be removed without losing any meaning from the study.

Response: We agree with the reviewer that the above sentence is speculative. Therefore, we removed the sentence from the text as recommended.

2. No changes in the manuscript are necessary, but just to respond to the rebuttal to point 7: Given the defective sucrose transport in the RNAi lines, it is difficult to conclude a direct correlation between decreased AGPase transcript abundance and decreased starch accumulation. The simplest explanation is that the reported decrease in starch content is related to the decreased sucrose transport. However, if the decrease in AGPase transcript abundance corresponds to the decrease in starch accumulation, the model becomes more complicated: Does this mean that the sucrose regulates AGPase at the transcriptional level (and the reduction in AGPase transcript is causative of the reduced starch) OR is the AGPase transcript level decreased due to some sort of feedback regulation from reduced starch accumulation (in which case the reduced sucrose supply is causative of the reduced starch content, and the AGPase transcript is a consequence). In their response, the authors seem to imply that it is the former - that reduced AGPase transcript that can explain the observed decrease in starch content. But in my opinion, there is not enough evidence to conclude one or the other.

Response: We appreciate the reviewer's notion about transcriptional regulation of AGPase, which is of course a very interesting point. AGPase is well known to be regulated at multiple levels which includes the tissue level of sucrose: high sucrose stimulates AGPase expression/activity. This can occur due to direct effects of sucrose on expression or rather indirect effects on enzyme activity via redox-modulation (Akihiro et al., 2005; Thiessen et al., 2002). In our transgenic caryopses we see reduced AGPase transcript abundance and at the same time reduced levels of sucrose. While this is just correlative (and not necessarily causative), it confirms existing knowledge. The present work does not allow making strict conclusions in this point; however, a valid hypothesis would be that lower flux of sucrose towards endosperm causes lower AGPase-expression/enzyme activity and this in turn leads to reduced levels of starch in the endosperm (This in turn affects sink activity and so on...). We aim to target this question in future investigations.

3. I noticed in Figure 1 the abbreviation AC for endosperm cavity while in S1 it was apoplastic cavity. Are these expressions used synonymously?

Response: Indeed, the terms „apoplastic cavity“ and „endosperm cavity“ are used synonymously. However, to avoid misunderstanding, we changed “endosperm cavity” to the “apoplastic cavity” in the legend to the Figure 1 to be in line with the remaining figures.